# HORIZON-FREE REINFORCEMENT LEARNING FOR LATENT MARKOV DECISION PROCESSES

## ABSTRACT

We study regret minimization for reinforcement learning (RL) in Latent Markov Decision Processes (LMDPs) with context in hindsight. We design a novel model-based algorithmic framework which can be instantiated with both a model-optimistic and a value-optimistic solver. We prove an $\widetilde{O}\left(\sqrt{M\Gamma SAK}\right)$ regret bound where $M$ is the number of contexts, $S$ is the number of states, $A$ is the number of actions, $K$ is the number of episodes, and $\Gamma \leqslant S$ is the maximum transition degree of any state-action pair. The regret bound only scales logarithmically with the planning horizon, thus yielding the first (nearly) horizon-free regret bound for LMDP. Key in our proof is an analysis of the total variance of alpha vectors, which is carefully bounded by a recursion-based technique. We complement our positive result with a novel $\Omega\left(\sqrt{MSAK}\right)$ regret lower bound with $\Gamma = 2$, which shows our upper bound minimax optimal when $\Gamma$ is a constant. Our lower bound relies on new constructions of hard instances and an argument based on the symmetrization technique from theoretical computer science, both of which are technically different from existing lower bound proof for MDPs, and thus can be of independent interest.

## 1 INTRODUCTION

One of the most popular model for Reinforcement Learning(RL) is Markov Decision Process (MDP), in which the transitions and rewards are dependent only on current state and agent's action. In standard MDPs, the agent has full observation of the state, so the optimal policy for the agent also only depends on states (called a *history-independent* policy). There is a line of research on MDPs, and the minimax regret and sample complexity guarantees have been derived.

Another popular model is Partially Observable MDPs (POMDPs) in which the agent only has partial observations of states. Even though the underlying transition is still Markovian, the lower bound for sample complexity has been proven to be exponential in state and action sizes. This is in part because the optimal policies for POMDPs are *history-dependent*.

In this paper we focus on a middle group between MDP and POMDP, namely Latent MDP (LMDP). An LMDP can be viewed as a collection of MDPs sharing the same state and action spaces, but the transitions and rewards may vary across them. Each MDP has a probability to be sampled at the beginning of each episode, and it will not change during the episode. The agent needs to find a policy which works well on these MDPs *in an average sense*. Empirically, LMDPs can be used for a wide variety of applications (Yu et al., 2020; Iakovleva et al., 2020; Finn et al., 2018; Ramamoorthy et al., 2013; Doshi-Velez & Konidaris, 2016; Yao et al., 2018). In general, there exists no policy that is optimally on every single MDP *simultaneously*, so this task is definitely harder than MDPs. On the other hand, LMDP is a special case of POMDP because for each MDP, the unobserved state is static in each episode and the observable state is just the state of MDP.

Unfortunately, for generic LMDPs, there exists exponential sample complexity lower bound (Kwon et al., 2021), so additional assumptions are needed to make the problem tractable. In this paper, we consider the setting that *after* each episode ends, the agent will get the context on which MDP it played with. Such information is often available. For example, in a maze navigation task, the location of the goal state can be viewed as the context.

In this setting, Kwon et al. (2021) obtained an $\widetilde{O}(\sqrt{MS^2AHK})$ regret upper bound where $M$ is the number of contexts, $S$ is the number of states, $A$ is the number of actions, $H$ is the planning horizon, and $K$ is the number of episodes. They did not study the regret lower bound.[1] To benchmark this result, the only available bound is $\widetilde{\Theta}\left(\sqrt{SAK}\right)$ from standard MDP by viewing MDP as a special case of LMDP.

Comparing these two bounds, we find significant gaps: ① Is the dependency on $M$ in LMDP necessary? ② The bound for MDP is (nearly) *horizon-free* (no dependency on $H$), is the polynomial dependency on $H$ in LMDP necessary? ③ The dependency on the number of states is $\sqrt{S}$ for MDP but the bound in Kwon et al. (2021) for LMDP is $S$.

In this paper, we resolve the first two questions and partially answer the third.

## 1.1 Main cntributions and technical novelties

We obtain the following new results:

• **Near-optimal regret guarantee for LMDPs.** We present an algorithm framework for LMDPs with context in hindsight. This framework can be instantiated with a plug-in solver for planning problems. We consider two types of solvers, one model-optimistic and one value-optimistic, and prove their regret bound to be $\widetilde{O}\left(\sqrt{M\Gamma SAK}\right)$ where $\Gamma \leqslant S$ is the maximum transition degree of any state-action pair. Compared with the result in Kwon et al. (2021), ours only requires the total reward to be bounded whereas they required a bounded reward for each step. Furthermore, we improve the $H$-dependence from $\sqrt{H}$ to logarithmic, making our bound (nearly) horizon-free. Lastly, our bound scales with $\sqrt{S\Gamma}$, which is strictly better than $S$ in their bound.

The main technique of our model-optimistic algorithm is to use a Bernstein-type confidence set on each position of transition dynamics, leading to a small Bellman error. The main difference between our value-optimistic algorithm and Kwon et al. (2021)'s is that we use a bonus depending on the variance of next-step values according to Bennett's inequality, instead of using Bernstein's inequality. It helps propagate the optimism from the last step to the first step, avoiding the $H$-dependency. We analyse these two solvers in a unified way, as their Bellman error are of the same order.

• **New regret lower bound for LMDPs.** We obtain a novel $\Omega\left(\sqrt{MSAK}\right)$ regret lower bound for LMDPs. This regret lower bound shows the dependency on $M$ is necessary for LMDPs. Notably the lower bound also implies $\widetilde{O}\left(\sqrt{M\Gamma SAK}\right)$ upper bound is optimal up to a $\sqrt{\Gamma}$ factor. Furthermore, our lower bound holds even for $\Gamma = 2$, which shows our upper bound is minimax optimal for a class of LMDPs with $\Gamma = O(1)$.

Our proof relies on new constructions of hard instances, different from existing ones for MDPs (Domingues et al., 2021). In particular, we use a two-phase structure to construct hard instances (cf. Figure 1). Furthermore, the previous approaches for proving lower bounds of MDPs do not work on LMDPs. For example, in the MDP instance of Domingues et al. (2021), the randomness comes from the algorithm and the last transition step before entering the good state or bad state. In an LMDP, the randomness of sampling the MDP from multiple MDPs must also be considered. Such randomness not only dilutes the value function by averaging over each MDP, but also divides the pushforward measure (see Page 3 of Domingues et al. (2021)) into $M$ parts. As a result, the $M$ terms in KL divergence in Equation (2) of Domingues et al. (2021) and that in Equation (10) cancels out — the final lower bound does not contain $M$. To overcome this, we adopt the *symmetrization technique* from theoretical computer science. This novel technique is helps generalize the bounds from a single-party result to a multiple-party result, which may give rise to a tighter lower bound.

---

[1]Their original bound is $\widetilde{O}(\sqrt{MS^2AH^3K})$ with the scaling that the reward from each step is bounded by 1. We rescale the reward to be bounded by $1/H$ in order to make the total reward from each episode bounded by 1, which is the setting we consider.

## 2 RELATED WORK

**LMDPs.**  As shown by Steimle et al. (2021), in the general cases, optimal policies for LMDPs are *history dependent* and P-SPACE hard to find. This is different from standard MDP cases where there always exists an optimal *history-independent* policy. However, even finding the optimal history-independent policy is *NP-hard* (Littman, 1994). Chades et al. (2012) provided heuristics for finding the optimal history-independent policy.

Kwon et al. (2021) investigated the sample complexity and regret bounds of LMDPs. Specifically, they presented an exponential lower-bound for general LMDPs without context in hindsign, and then they derived an algorithm with polynomial sample complexity and sub-linear regret for two special cases (with context in hindsight, or $\delta$-strongly separated MDPs).

LMDP has been studied as a type of multi-task RL (Taylor & Stone, 2009; Brunskill & Li, 2013; Liu et al., 2016; Hallak et al., 2015). It has been applied to model combinatorial optimization problems (Zhou et al., 2022). There are also some related studies such as model transfer (Lazaric, 2012; Zhang & Wang, 2021) and contextual decision processes (Jiang et al., 2017). In empirical works, LMDP has has wide applications in multi-task RL (Yu et al., 2020), meta RL Iakovleva et al. (2020); Finn et al. (2018), latent-variable MDPs (Ramamoorthy et al., 2013) and hidden parameter MDPs (Doshi-Velez & Konidaris, 2016; Yao et al., 2018).

**Regret Analysis for MDPs.**  LMDPs are generalizations of MDPs, so some previous approaches to solving MDPs can provide insights. There is a long line of work on regret analysis for MDPs  (Azar et al., 2017; Dann et al., 2017; 2019; Zanette & Brunskill, 2019; Zhang et al., 2021a). In this paper, we focus on time-homogeneous, finite horizon, undiscounted MDPs whose total reward is upper-bounded by 1. Recent work showed in this setting the regret can be (nearly) horizon-free for tabular MDPs Wang et al. (2020); Zhang et al. (2022; 2021a; 2020); Ren et al. (2021). Importantly these results indicate RL may not be more difficult than bandits in the minimax sense. More recent work generalized the horizon-free results to other MDP problems (Zhang et al., 2021b; Kim et al., 2021; Tarbouriech et al., 2021; Zhou & Gu, 2022). However, all existing work with horizon-free guarantees only considered single-environment problems. Ours is the first horizon-free guarantee that goes beyond MDP.

Neu & Pike-Burke (2020) summarized up the "optimism in the face of uncertainty" (OFU) principle in RL. They named two types of optimism: ① *model-optimistic* algorithms construct confidence sets around empirical transitions and rewards, and select the policy with the highest value in the best possible models in these sets. ② *value-optimistic* algorithms construct upper bounds on the optimal value functions, and select the policy which maximizes this optimistic value function. Our paper follows their idea and provide one algorithm for each type of optimism.

## 3 PROBLEM SETUP

In this section, we give a formal definition of Latent Markov Decision Processes (Latent MDPs).

**Notations.**  For any event $\mathcal{E}$, we use $\mathbb{1}[\mathcal{E}]$ to denote the indicator function, i.e., $\mathbb{1}[\mathcal{E}] = 1$ if $\mathcal{E}$ holds and $\mathbb{1}[\mathcal{E}] = 0$ otherwise. For any set $X$, we use $\Delta(X)$ to denote the probability simplex over $X$. For any positive integer $n$, we use $[n]$ to denote the set $\{1, 2, \ldots, n\}$. For any probability distribution $P$, we use $\mathsf{supp}(P) = \|P\|_0$ to denote the size of support of $P$, i.e., $\sum_x \mathbb{1}[P(x) > 0]$. There are three ways to denote a $d$-dimensional vector (function): suppose $p$ is any parameter, $x(p) = (x_1(p), x_2(p), \ldots, x_d(p))$ if the indices are natural numbers, $x(\cdot|p) = (x(i_1|p), x(i_2|p), \ldots, x(i_d|p))$ and $x(p\cdot) = (x(pi_1), x(pi_2), \ldots, x(pi_d))$ if the indices are from the set $I = \{i_1, i_2, \ldots, i_d\}$. For any number $q$, we use $x^q$ to denote the vector $(x_1^q, x_2^q, \ldots, x_d^q)$. For two $d$-dimensional vectors $x$ and $y$, we use $x^\top y = \sum_i x_i y_i$ to denote the inner product. If $x$ is a probability distribution, we use $\mathbb{V}(x, y) = \sum_i x_i(y_i - x^\top y)^2 = x^\top(y^2) - (x^\top y)^2$ to denote the empirical variance. We use $\iota = 2\ln\left(\frac{2MSAHK}{\delta}\right)$ as a log term where $\delta$ is the confidence parameter.

## 3.1 Latent Markov Decision Process

Latent MDP (Kwon et al., 2021) is a collection of finitely many MDPs $\mathcal{M} = \{\mathcal{M}_1, \mathcal{M}_2, \ldots, \mathcal{M}_M\}$ where $M = |\mathcal{M}|$. All the MDPs share state set $\mathcal{S}$, action set $\mathcal{A}$ and horizon $H$. Each MDP $\mathcal{M}_m = (\mathcal{S}, \mathcal{A}, H, \nu_m, P_m, R_m)$ has its own initial state distribution $\nu_m \in \Delta(\mathcal{S})$, transition model $P_m : \mathcal{S} \times \mathcal{A} \to \Delta(\mathcal{S})$ and a deterministic reward function $R_m : \mathcal{S} \times \mathcal{A} \to [0, 1]$. Let $w_1, w_2, \ldots, w_M$ be the mixing weights of MDPs such that $w_m > 0$ for any $m$ and $\sum_{m=1}^{M} w_m = 1$.

Denote $S = |\mathcal{S}|, A = |\mathcal{A}|$ and $\Gamma = \max_{m,s,a} \mathsf{supp}\left(P_m(\cdot|s,a)\right)$. $\Gamma$ can be interpreted as the maximum degree of each transition, which is a quantity our regret bound depends on. Note we always have $\Gamma \leqslant S$. In previous work, Lattimore & Hutter (2012) assumes $\Gamma = 2$, and Fruit et al. (2020) also has a regret bound that scales with $\Gamma$.

In the worst case, the optimal policy of an LMDP is history-dependent and is PSPACE-hard to find (Corollary 1 and Proposition 3 in Steimle et al. (2021)). Aside from computational difficulty, storing a history-dependent policy needs a space which is exponentially large, so it is generally impractical. In this paper, we seek to provide a result for any fixed policy class $\Pi$. For example, we can have $\Pi$ to be the set of all history-independent, deterministic policies to alleviate the space issue. Following previous work (Kwon et al., 2021), we assume access to oracles for planning and optimization. See Section 4 for the formal definitions.

We consider an episodic, finite-horizon and undiscounted reinforcement learning problem on LMDPs. In this problem, the agent interacts with the environment for $K$ episodes. At the start of every episode, one MDP $\mathcal{M}_m \in \mathcal{M}$ is randomly chosen with probability $w_m$. Throughout the episode, the true context is *hidden*. The agent can only choose actions based on the history information up until current time. However, at the end of each episode (after $H$ steps), the agent gets revealed the true context $m$. This permits an unbiased model estimation for the LMDP. As in Cohen et al. (2020), the central difficulty is to estimate the transition, we also focus on learning $P$ only. For simplicity, we assume that $w, \nu$ and $R$ are *known* to the agent, because they can be estimated easily. The assumption of deterministic rewards is also for simplicity. Our analysis can be extended to unknown and bounded-support reward distributions.

## 3.2 Value functions, Q-functions and alpha vectors

By convention, the expected reward of executing a policy on any MDP can be defined via value function $V$ and Q-function $Q$. Since for MDPs there is always an optimal policy which is history-independent, $V$ and $Q$ only need the current state and action as parameters.

However, these notations fall short of history-independent policies under the LMDP setting. The full information is encoded in the *history*, so here we use a more generalized definition called *alpha vector* (following notations in Kwon et al. (2021)). For any time $t \geqslant 1$, let $h_t = (s, a, r)_{1:t-1} s_t$ be the history up until time $t$. Define $\mathcal{H}_t$ as the set of histories observable at time step $t$, and $\mathcal{H} := \cup_{t=1}^{H} \mathcal{H}_t$ as the set of all possible histories. We define the alpha vectors $\alpha_m^\pi(h)$ for $(m, h) \in [M] \times \mathcal{H}$ as follows:

$$\alpha_m^\pi(h) := \mathbb{E}\left[ \sum_{t'=t}^{H} R_m(s_{t'}, a_{t'}) \,\middle|\, \mathcal{M}_m, \pi, h_t = h \right],$$

$$\alpha_m^\pi(h, a) := \mathbb{E}\left[ \sum_{t'=t}^{H} R_m(s_{t'}, a_{t'}) \,\middle|\, \mathcal{M}_m, \pi, (h_t, a_t) = (h, a) \right].$$

The alpha vectors are indeed value functions and Q-functions on each individual MDP.

Next, we introduce the concepts of belief state to show how to do planning in LMDP. Let $b_m(h)$ denote the belief state over $M$ MDPs corresponding to a history $h$, i.e., the probability of the true MDP being $\mathcal{M}_m$ conditioned on observing history $h$. We have the following recursion:

$$b_m(s) = \frac{w_m \nu_m(s)}{\sum_{m'=1}^{M} w_{m'} \nu_{m'}(s)} \quad \text{and} \quad b_m(hars') = \frac{b_m(h) P_m(s'|s,a) \mathbb{1}[r = R_m(s,a)]}{\sum_{m'=1}^{M} b_{m'}(h) P_{m'}(s'|s,a) \mathbb{1}[r = R_{m'}(s,a)]}.$$

The value functions and Q-functions for LMDP is defined via belief states and alpha vectors:

$$V^\pi(h) := b(h)^\top \alpha^\pi(h) \quad \text{and} \quad Q^\pi(h, a) := b(h)^\top \alpha^\pi(h, a).$$

Direct computation (see Appendix B.1) gives

$$V^\pi(h) = \sum_{a\in\mathcal{A}} \pi(a|h) \underbrace{\left( b(h)^\top R(s,a) + \sum_{s'\in\mathcal{S},r} \sum_{m'=1}^{M} b_{m'}(h) P_{m'}(s'|s,a) \mathbb{1}[r = R_{m'}(s,a)] V^\pi(hars') \right)}_{=Q^\pi(h,a)}.$$

So planning in LMDP can be viewed as planning in belief states. For the optimal *history-dependent* policy, we can select

$$\pi(h) = \arg\max_{a\in\mathcal{A}} \left( b(h)^\top R(s,a) + \sum_{s'\in\mathcal{S},r} \sum_{m'=1}^{M} b_{m'}(h) P_{m'}(s'|s,a) \mathbb{1}[r = R_{m'}(s,a)] V^\pi(hars') \right), \quad (1)$$

using dynamic programming in descending order of $h$'s length.

### 3.3 PERFORMANCE MEASURE.

We use cumulative regret to measure the algorithm's performance. The optimal policy is $\pi^\star = \arg\max_{\pi\in\Pi} V^\pi$, which also does not know the context when interacting with the LMDP. Suppose the agent interacts with the environment for $K$ episodes, and for each episode $k$ a policy $\pi^k$ is played. The regret is defined as

$$\mathbf{Regret}(K) := KV^\star - \sum_{k=1}^{K} V^{\pi^k}.$$

## 4 MAIN ALGORITHMS AND RESULTS

In this section, we present two algorithms, and show their minimax regret guarantee. The first is to use a Bernstein confidence set on transition probabilities, which was first applied to SSP in Cohen et al. (2020) to derive a horizon-free regret. This algorithm uses a bi-level optimization oracle: for the inner layer, an oracle is needed to find the optimal policy inside $\Pi$ under a given LMDP; for the outer layer, an oracle finds the best transition inside the confidence set which maximizes the optimal expected reward. The second is to adapt the Monotoic Value Propagation (MVP) algorithm (Zhang et al., 2021a) to LMDPs. This algorithm requires an oracle to solve an LMDP with *dynamic bonus*: the bonuses depends on the variances of the next-step alpha vector. Both algorithms enjoy the following regret guarantee.

**Theorem 1.** *For both the Bernstein confidence set for LMDP (Algorithm 1 combined with Algorithm 2) and the Monotonic Value Propagation for LMDP (Algorithm 1 combined with Algorithm 3), with probability at least $1 - \delta$, we have that*

$$\mathbf{Regret}(K) = O\left( \sqrt{M\Gamma SAK} \ln\left(\frac{MSAHK}{\delta}\right) + MS^2 A \ln^2\left(\frac{MSAHK}{\delta}\right) \right)$$

As we have discussed, our result improves upen Kwon et al. (2021), and has only logarithmic dependency on the planning horizon $H$. We also have a lower order which scales with $S^2$. We note that even in the standard MDP setting, it remains a major open problem how to obtain minimax optimal regret bound with no lower order term (Zhang et al., 2021a).

Below we describe the details of our algorithms.

**Algorithm framework.** The two algorithms introduced in this section share a framework for estimating the model. The only difference between them is the solver for the exploration policy. The framework is shown in Algorithm 1. Our algorithmic framework estimates the model (cf. Line 14 in Algorithm 1) and then selects a policy for the next round based on different oracles (cf. Line 18 in Algorithm 1). Following Zhang et al. (2021a), we use a doubling schedule for each state-action pair in every MDP to update the estimation and exploration policy.

**Common notations.** Some of the notations have been introduced in Algorithm 1, but for reading convenience we will repeat the notations here. For any notation, we put the episode number $k$ in the superscript. For any observation, we put the time step $t$ in the subscript. For any model component, we put the context $m$ in the subscript. The alpha vector and value function for the optimistic model are denoted using an extra "~".

**Algorithm 1** Algorithmic Framework for Solving LMDPs

---

1: **Input:** Number of MDPs $M$, state space $\mathcal{S}$, action space $\mathcal{A}$, horizon $H$; policy class $\Pi$; confidence parameter $\delta$.
2: Set an arbitrary policy $\pi^1$, initialize all $n, N$ with 0 and set constant $\iota \leftarrow 2\ln\left(\frac{2MSAHK}{\delta}\right)$.
3: **for** $k = 1, 2, \ldots, K$ **do**
4:    **for** $t = 1, 2, \ldots, H$ **do**
5:       Observe state $s_t^k$.
6:       Choose action $a_t^k = \pi^k(s_t^k)$.
7:    **end for**
8:    Observe state $s_{H+1}^k$ and get $m^k$ as hindsight.
9:    **for** $t = 1, 2, \ldots, H$ **do**
10:       Set $n_{m^k}(s_t^k, a_t^k) \leftarrow n_{m^k}(s_t^k, a_t^k) + 1$ and $n_{m^k}(s_{t+1}^k|s_t^k, a_t^k) \leftarrow n_{m^k}(s_{t+1}^k|s_t^k, a_t^k) + 1$.
11:       **if** $\exists i \in \mathbb{N}, n_{m^k}(s_t^k, a_t^k) = 2^i$ **then**
12:          Set TRIGGERED = TRUE.
13:          Set $N_{m^k}(s_t^k, a_t^k) \leftarrow n_{m^k}(s_t^k, a_t^k)$.
14:          Set $\widehat{P}_m(s'|s_t^k, a_t^k) \leftarrow \frac{n_{m^k}(s'|s_t^k, a_t^k)}{n_{m^k}(s_t^k, a_t^k)}$ for all $s' \in \mathcal{S}$.
15:       **end if**
16:    **end for**
17:    **if** TRIGGERED **then**
18:       Set $\pi^{k+1} \leftarrow$ `Solver()` (by Algorithm 2 or Algorithm 3).
19:    **else**
20:       Set $\pi^{k+1} \leftarrow \pi^k$.
21:    **end if**
22: **end for**

---

### 4.1 Bernstein confidence set of transitions for LMDPs

We introduce a model-optimistic approach by using a confidence set of transition probability.

**Optimistic LMDP construction.** The Bernstein confidence set is constructed as below:

$$\mathcal{P}^{k+1} = \left\{ \widetilde{P} : \left|\left(\widetilde{P}_m - \widehat{P}_m^k\right)(s'|s,a)\right| \leqslant 2\sqrt{\frac{\widehat{P}_m^k(s'|s,a)\iota}{N_m^k(s,a)}} + \frac{5\iota}{N_m^k(s,a)}, \forall (m,s,a,s) \in [M] \times \mathcal{S} \times \mathcal{A} \times \mathcal{S} \right\}. \quad (2)$$

Notice that we do not change the reward function, so we still have the total reward of any trajectory upper-bounded by 1.

**Policy solver.** The policy solver is in Algorithm 2. It solves a two-step optimization problem on Line 2: for the inner problem, given a transition model $\widetilde{P}$ and all other known quantities $w, \nu, R$, it needs a planning oracle to find the optimal policy; for the outer problem, it needs to find the optimal transition model. For planning, we can use the method presented in Equation (1). For the outer problem, we can use Extended Value Iteration as in Auer et al. (2008); Fruit et al. (2020); Filippi et al. (2010); Cohen et al. (2020). For notational convenience, we denote the alpha vectors and value functions calculated by $\widetilde{P}^k$ and $\pi^k$ as $\widetilde{\alpha}^k$ and $\widetilde{V}^k$.

---

**Algorithm 2** `Solver-L-Bernstein`

---

1: Construct $\mathcal{P}^{k+1}$ using Equation (2).
2: Find $\widetilde{P}^{k+1} \leftarrow \arg\max_{\widetilde{P} \in \mathcal{P}^{k+1}} \left(\max_{\pi \in \Pi} V_{\widetilde{P}}^\pi\right)$.
3: Find $\pi^{k+1} \leftarrow \arg\max_{\pi \in \Pi} V_{\widetilde{P}^{k+1}}^\pi$.
4: **Return:** $\pi^{k+1}$.

---

### 4.2 Monotonic Value Propagation for LMDP

We introduce a value-optimistic approach by calculating a variance-dependent bonus. This technique was originally used to solve standard MDPs (Zhang et al., 2021a).

**Optimistic LMDP construction.** The optimistic model contains a bonus function, which is inductively defined using the next-step alpha vector. In episode $k$, or any policy $\pi$, assume the alpha vector for any history with length $t+1$ is calculated, then for any history $h$ with length $t$, the bonus is defined as follows:

$$B_m^k(h,a) := \max\left\{4\sqrt{\frac{\mathsf{supp}\left(\widehat{P}_m^k(\cdot|s,a)\right)\mathbb{V}\left(\widehat{P}_m^k(\cdot|s,a),\widetilde{\alpha}_m^\pi(har\cdot)\right)\iota}{N_m^k(s,a)}},\ \frac{16S\iota}{N_m^k(s,a)}\right\},\quad (3)$$

where $r = R_m(s,a)$. Next, the alpha vector of history $h$ is:

$$\widetilde{\alpha}_m^\pi(h) := \min\left\{R_m(s,a) + B_m^k(h,a) + \widehat{P}_m^k(\cdot|s,a)^\top\widetilde{\alpha}_m^\pi(har\cdot),\ 1\right\},\ \text{where } a = \pi(h).\quad (4)$$

Finally, the value function is:

$$\widetilde{V}^\pi := \sum_{m=1}^M\sum_{s\in\mathcal{S}}w_m\nu_m(s)\widetilde{\alpha}_m^\pi(s).\quad (5)$$

**Policy solver.** The policy solver is in Algorithm 3. It finds the policy maximizing the optimistic value, with a dynamic bonus function depending on the policy itself. This solver is tractable if we only care about *deterministic* policies in $\Pi$. This restriction is reasonable because for the original LMDP there always exists an optimal policy which is deterministic. Further, according to the proof of Lemma 13, we only need a policy which has optimistic value no less than that of the optimal value. Thus, there always exists an exhaustive search algorithm for this solver, which enumerates each action at each history.

---

**Algorithm 3** `Solver-L-MVP`

---
1: Use the optimistic model defined in Equation (3), Equation (4) and Equation (5).
2: Find $\pi^{k+1} \leftarrow \arg\max_{\pi\in\Pi}\widetilde{V}^\pi$.
3: **Return:** $\pi^{k+1}$.

---

## 5 REGRET LOWER BOUND

In this section, we present a regret lower bound for the unconstrained policy class, i.e., when $\Pi$ contains all possible history-dependent policies.

First, we note that this lower bound cannot be directly reduced to solving $M$ MDPs (with the context revealed at the beginning of each episode). Because simply changing the time of revealing the context results in the change of the optimal policy and its value function.

At a high level, our approach is to transform the problem of context in hindsight into a problem of essentially context being told beforehand, while *not affecting the optimal value function*. To achieve this, we can use a small portion of states to encode the context at the beginning, then the optimal policy can extract information from them and fully determine the context.

After the transformation, we can view the LMDP as a set of independent MDPs, so it is natural to leverage results from MDP lower bounds. Intuitively, since the lower bound of MDP is $\sqrt{SAK}$, and each MDP is assigned roughly $\frac{K}{M}$ episodes, the lower bound of LMDP is $M\sqrt{SA\cdot\frac{K}{M}} = \sqrt{MSAK}$. To formally prove this, we adopt the *symmetrization technique* from the theoretical computer science community (Phillips et al., 2012; Woodruff & Zhang, 2014; Fischer et al., 2017; Vempala et al., 2020). When an algorithm interacts with an LMDP, we can focus on each MDP, while viewing the interactions with other MDPs as irrelevant – we hard code the other MDPs into the algorithm, deriving an algorithm for an *MDP*. In other words, we can insert an MDP into any of the $M$ positions, and they are all symmetric to the algorithm's view. So, the regret can be averagely distributed to each MDP.

The main theorem is shown here, before we introduce the construction of LMDP instances. Its proof is placed in Appendix B.4.

**Theorem 2.** *Assume that $S \geqslant 6$, $A \geqslant 2$ and $M \leqslant \lfloor \frac{S}{2} \rfloor!$. For any algorithm $\boldsymbol{\pi}$, there exists an LMDP $\mathcal{M}_{\boldsymbol{\pi}}$ such that, for $K \geqslant \widetilde{\Omega}(M^2 + MSA)$, its expected regret in $\mathcal{M}_{\boldsymbol{\pi}}$ after $K$ episodes satisfies*

$$R(\mathcal{M}_{\boldsymbol{\pi}}, \boldsymbol{\pi}, K) := \mathbb{E}\left[ \sum_{k=1}^{K} (V^{\star} - V^k) \,\middle|\, \mathcal{M}_{\boldsymbol{\pi}}, \boldsymbol{\pi} \right] = \Omega\left( \sqrt{MSAK} \right).$$

Several remarks are in the sequel. ① This is the first regret lower bound for LMDPs with context in hindsight. To the best of our knowledge, the introduction of the symmetrization technique is novel to the construction of lower bounds in the field of RL. ② This lower bound matches the minimax regret upper bound (Theorem 1) up to logarithm factors, because in the hard instance construction $\Gamma = 2$. For general cases, our upper bound is optimal up to a $\sqrt{\Gamma}$ factor. ③ There is a limitation of $M$, which could be at most $\lfloor \frac{S}{2} \rfloor!$, though an exponentially large $M$ is not practical.

## 5.1 Hard instance construction

Since $M \leqslant \lfloor \frac{S}{2} \rfloor!$, we can always find an integer $d_1$ such that $d_1 \leqslant \frac{S}{2}$ and $M \leqslant d_1!$. Since $S \geqslant 6$ and $d_1 \leqslant \frac{S}{2}$, we can always find an integer $d_2$ such that $d_2 \geqslant 1$ and $2^{d_2} - 1 \leqslant S - d_1 - 2 < 2^{d_2+1} - 1$. We can construct a two-phase structure, each phase containing $d_1$ and $d_2$ steps respectively.

The hard instance uses similar components as the MDP instances in Domingues et al. (2021). We construct *a collection of LMDPs* $\mathcal{C} := \{\mathcal{M}_{(\boldsymbol{\ell}^{\star}, \boldsymbol{a}^{\star})} : (\boldsymbol{\ell}^{\star}, \boldsymbol{a}^{\star}) \in [L]^M \times [A]^M\}$, where we define $L := 2^{d_2-1} = \Theta(S)$. For a fixed pair $(\boldsymbol{\ell}^{\star}, \boldsymbol{a}^{\star}) = ((\ell_1^{\star}, \ell_2^{\star}, \ldots, \ell_m^{\star}), (a_1^{\star}, a_2^{\star}, \ldots, a_m^{\star}))$, we construct the LMDP $\mathcal{M}_{(\boldsymbol{\ell}^{\star}, \boldsymbol{a}^{\star})}$ as follows.

### 5.1.1 The LMDP layout

All MDPs in the LMDP share the same logical structure. Each MDP contains two phases: the encoding phase and the guessing phase. The encoding phase contains $d_1$ states, sufficient for encoding the context because $M \leqslant d_1!$. The guessing phase contains a number guessing game with $C := LA = \Theta(SA)$ choices. If the agent makes a correct choice, it receives an expected reward slightly larger than $\frac{1}{2}$. Otherwise, it receives an an expected reward of $\frac{1}{2}$.

### 5.1.2 The detailed model

Now we give more details about our construction. Figure 1 shows an example of the model with $M = 2$, $S = 11$, arbitrary $A \geqslant 2$ and $H \geqslant 6$.

**States.** The states in the encoding phase are $e_1, \ldots, e_{d_1}$. The states in the guessing phase are $s_1, \ldots, s_N$ where $N = \sum_{i=0}^{d_2-1} 2^i = 2^{d_2} - 1$. There is a good state $g$ for reward and a terminal state $t$. All the unused states can be ignored.

**Transitions.** The weights are equal, i.e., $w_m = \frac{1}{M}$. We assign a unique integer in $m \in [M]$ to each MDP as a context. Each integer $m$ is uniquely mapped to a permutation $\boldsymbol{\sigma}(m) = (\sigma_1(m), \sigma_2(m), \ldots, \sigma_{d_1}(m))$. Then the initial state distribution is $\nu_m(e_{\sigma_1(m)}) = 1$. The transitions for the first $d_1$ steps are: for any $(m, a) \in [M] \times \mathcal{A}$,

$$P_m(e_{\sigma_{i+1}(m)} \mid e_{\sigma_i(m)}, a) = 1, \ \forall 1 \leqslant i \leqslant d_1 - 1; \quad P_m(s_1 \mid e_{\sigma_{d_1}(m)}, a) = 1.$$

This means, in the encoding phase, whatever the agent does is irrelevant to the state sequence it observes.

The guessing phase is a binary tree which we modify from Section 3.1 of Domingues et al. (2021) (here we equal each action $a$ to an integer in $[A]$): for any $(m, a) \in [M] \times \mathcal{A}$,

$$P_m(s_{2i+(a \bmod 2)} \mid s_i, a) = 1, \ \forall 1 \leqslant i \leqslant 2^{d_2-1} - 1.$$

For the tree leaves $\mathcal{L} = \{s_\ell : 2^{d_2-1} \leqslant \ell \leqslant 2^{d_2} - 1\}$ (notice that $|\mathcal{L}| = L$), we construct: for any $(m, \ell, a) \in [M] \times \mathcal{L} \times \mathcal{A}$,

$$P_m(t \mid s_\ell, a) = \frac{1}{2} - \varepsilon \mathbb{1}[\ell = \ell_m^{\star}, a = a_m^{\star}], \quad P_m(g \mid s_\ell, a) = \frac{1}{2} + \varepsilon \mathbb{1}[\ell = \ell_m^{\star}, a = a_m^{\star}].$$

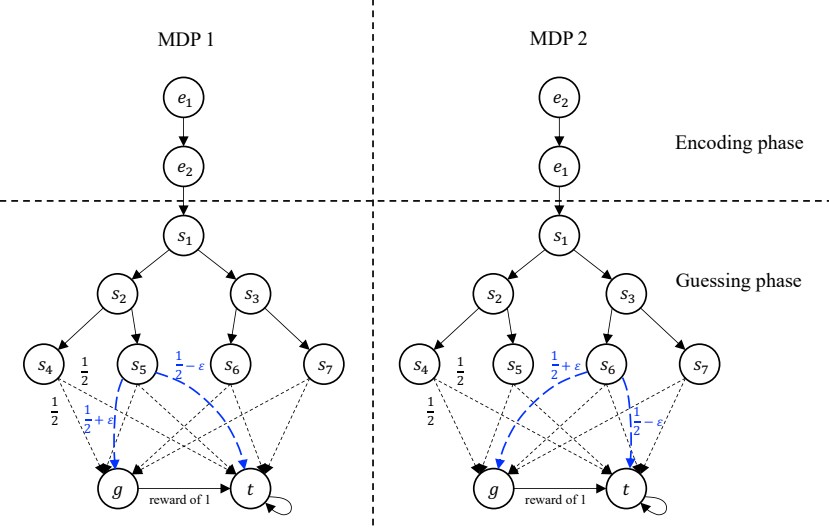

**Figure 1:** Illustration of the class of hard LMDPs used in the proof of Theorem 2. Solid arrows are deterministic transitions, while dashed arrows are probabilistic transitions. The probabilities are written aside of the transitions. For any of the MDP, the agent first goes through an encoding phase, where it observes a sequence of states regardless of what actions it take. The state sequence is different for each MDP, so the agent can fully determine which context it is in after this phase. When in the guessing phase, the agent needs to travel through a binary tree until it gets to some leaf. Exactly one of the leaves are "correct", and only performing exactly one of the actions at the correct leaf yields an expected higher reward.

Recall that we denote $C = LA$ as the effective number of choices. The agent needs to first find the correct leaf by inputting its binary representation correctly, then choose the correct action.

The good state is temporary between the guessing phase and the terminal state: if the agent is at $g$ and makes any action, it enters $t$. The terminal state is self-absorbing. For any $(m, a) \in [M] \times \mathcal{A}$,

$$P_m(t \mid g, a) = 1, \ P_m(t \mid t, a) = 1.$$

All the unmentioned probabilities are $0$. Clearly, this transition model guarantees that $\mathsf{supp}(P_m(\cdot | s, a)) \leqslant 2$ for any pair of $(m, s, a) \in [M] \times \mathcal{S} \times \mathcal{A}$.

**The rewards.** The only non-zero rewards are $R_m(g, a) = 1$ for any $(m, a) \in [M] \times \mathcal{A}$. Since this state-action pair is visited at most once in any episode, this reward guarantees that in a single episode the cumulative reward is either $0$ or $1$.

## 6 CONCLUSION

In this paper, we present two different RL algorithms (one model-optimistic and one value-optimistic) for LMDPs with context in hindsight, both achieving $\widetilde{O}(\sqrt{M\Gamma SAK})$ regret. This is the first (nearly) horizon-free regret bound for LMDP with context in hindsight. We also provide a regret lower bound for this setting, which is $\Omega(\sqrt{MSAK})$. In this lower bound, $\Gamma = 2$, so the upper bound is minimax optimal for the subclass of LMDPs with constant $\Gamma$. One future direction is to obtain a minimax regret bound for LMDPs for the $\Gamma = \Theta(S)$ case. For example, can we derive a regret lower bound of $\Omega(\sqrt{MS^2 AK})$? On the other hand, it is also possible to remove the $\sqrt{\Gamma}$ in our upper bound. We believe this will require properties beyond the standard Bellman-optimality condition for standard MDPs.

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

## A  TECHNICAL LEMMAS

**Lemma 3** (Anytime Azuma, Theorem D.1 in Cohen et al. (2020)). *Let $(X_n)_{n=1}^{\infty}$ be a martingale difference sequence with respect to the filtration $(\mathcal{F}_n)_{n=0}^{\infty}$ such that $|X_n| \leqslant B$ almost surely. Then with probability at least $1 - \delta$,*

$$\left| \sum_{i=1}^{n} X_i \right| \leqslant B \sqrt{n \ln \frac{2n}{\delta}}, \quad \forall n \geqslant 1.$$

**Lemma 4** (Bennett's Inequality, Theorem 3 in Maurer & Pontil (2009)). *Let $Z, Z_1, \ldots, Z_n$ be i.i.d. random variables with values in $[0, b]$ and let $\delta > 0$. Define $\mathbb{V}[Z] = \mathbb{E}[(Z - \mathbb{E}[Z])^2]$. Then we have*

$$\mathbb{P}\left[ \left| \mathbb{E}[Z] - \frac{1}{n}\sum_{i=1}^{n} Z_i \right| > \sqrt{\frac{2\mathbb{V}[Z]\ln(2/\delta)}{n}} + \frac{b\ln(2/\delta)}{n} \right] \leqslant \delta.$$

**Lemma 5** (Theorem 4 in Maurer & Pontil (2009)). *Let $Z, Z_1, \ldots, Z_n$ $(n \geqslant 2)$ be i.i.d. random variables with values in $[0, b]$ and let $\delta > 0$. Define $\bar{Z} = \frac{1}{n}Z_i$ and $\hat{V}_n = \frac{1}{n}\sum_{i=1}^{n}(Z_i - \bar{Z})^2$. Then we have*

$$\mathbb{P}\left[ \left| \mathbb{E}[Z] - \frac{1}{n}\sum_{i=1}^{n} Z_i \right| > \sqrt{\frac{2\hat{V}_n \ln(2/\delta)}{n-1}} + \frac{7b\ln(2/\delta)}{3(n-1)} \right] \leqslant \delta.$$

**Lemma 6** (Lemma 30 in Tarbouriech et al. (2021)). *Let $(M_n)_{n \geqslant 0}$ be a martingale such that $M_0 = 0$ and $|M_n - M_{n-1}| \leqslant c$ for some $c > 0$ and any $n \geqslant 1$. Let $\mathrm{Var}_n = \sum_{k=1}^{n} \mathbb{E}[(M_k - M_{k-1})^2 | \mathcal{F}_{k-1}]$ for $n \geqslant 0$, where $\mathcal{F}_k = \sigma(X_4, \ldots, M_k)$. Then for any positive integer $n$ and $\delta \in (0, 2(nc^2)^{1/\ln 2}]$, we have that*

$$\mathbb{P}\left[ |M_n| \geqslant 2\sqrt{2\mathrm{Var}_n(\log_2(nc^2) + \ln(2/\delta))} + 2\sqrt{\log_2(nc^2) + \ln(2/\delta)} + 2c(\log_2(nc^2) + \ln(2/\delta)) \right] \leqslant \delta.$$

**Lemma 7** (Lemma 11 in Zhang et al. (2021a)). *Let $\lambda_1, \lambda_2, \lambda_4 \geqslant 0$, $\lambda_3 \geqslant 1$ and $i' = \log_2 \lambda_1$. Let $a_1, a_2, \ldots, a_{i'}$ be non-negative reals such that $a_i \leqslant \lambda_1$ and $a_i \leqslant \lambda_2\sqrt{a_{i+1} + 2^{i+1}\lambda_3} + \lambda_4$ for any $1 \leqslant i \leqslant i'$. Then we have that $a_1 \leqslant \max\{(\lambda_2 + \sqrt{\lambda_2^2 + \lambda_4})^2, \lambda_2\sqrt{8\lambda_3} + \lambda_4\}$.*

## B  SKIPPED PROOFS

### B.1  OMITTED CALCULATION

Here we give the details for ommitted calculations.

$$
\begin{aligned}
V^{\pi}(h) &= \sum_{a \in \mathcal{A}} \pi(a|h) \left( b(h)^{\top} R(s,a) + \sum_{s' \in \mathcal{S}, r} \sum_{m=1}^{M} b_m(h) P_m(s'|s,a) \mathbb{1}[r = R_m(s,a)] \alpha_m^{\pi}(hars') \right) \\
&= \sum_{a \in \mathcal{A}} \pi(a|h) \left( b(h)^{\top} R(s,a) + \sum_{s' \in \mathcal{S}, r} \sum_{m'=1}^{M} b_{m'}(h) P_{m'}(s'|s,a) \mathbb{1}[r = R_{m'}(s,a)] \sum_{m=1}^{M} b_m(hars') \alpha_m^{\pi}(hars') \right) \\
&= \sum_{a \in \mathcal{A}} \pi(a|h) \underbrace{\left( b(h)^{\top} R(s,a) + \sum_{s' \in \mathcal{S}, r} \sum_{m'=1}^{M} b_{m'}(h) P_{m'}(s'|s,a) \mathbb{1}[r = R_{m'}(s,a)] V^{\pi}(hars') \right)}_{=Q^{\pi}(h,a)}.
\end{aligned}
$$

### B.2  UNIFIED ANALYSES OF ALGORITHM 1, ALGORITHM 2 AND ALGORITHM 3

In this subsection, we present the proof of Theorem 1 by showing each step. However, when encountered with some lemmas, the proofs of lemmas are skipped and deferred to Appendix B.3.

**Good events.** The entire proof depends heavily on the good events defined below in Definition 8. They show that the estimation of transition probability is very close to the true value. We show in Lemma 9 that they happen with a high probability.

**Definition 8** (Good events). *For every episode $k$, define the following events:*

$$\Omega_1^k := \left\{ \forall (m,s,a,s') \in [M] \times \mathcal{S} \times \mathcal{A} \times \mathcal{S}, \; \left| \left( \widehat{P}_m^k - P_m \right)(s'|s,a) \right| \leqslant 2\sqrt{\frac{\widehat{P}_m^k(s'|s,a)\iota}{N_m^k(s,a)}} + \frac{5\iota}{N_m^k(s,a)} \right\}, \quad (6)$$

$$\Omega_2^k := \left\{ \forall (m,s,a,s') \in [M] \times \mathcal{S} \times \mathcal{A} \times \mathcal{S}, \; \left| \left( \widehat{P}_m^k - P_m \right)(s'|s,a) \right| \leqslant \sqrt{\frac{2P_m(s'|s,a)\iota}{N_m^k(s,a)}} + \frac{\iota}{N_m^k(s,a)} \right\}. \quad (7)$$

*Further, define $\Omega_1 := \cap_{k=1}^K \Omega_1^k$ and $\Omega_2 := \cap_{k=1}^K \Omega_2^k$.*

**Lemma 9.** $\mathbb{P}[\Omega_1], \mathbb{P}[\Omega_2] \geqslant 1 - \delta.$

Assume that good events hold, then we have the following useful property:

**Lemma 10.** *Conditioned on $\Omega_1$, we have that for any $(m,s,a,k) \in [M] \times \mathcal{S} \times \mathcal{A} \times [K]$, and any $S$-dimensional vector $\alpha$ such that $\|\alpha\|_\infty \leqslant 1$,*

$$\left| \left( \widehat{P}_m^k - P_m \right)(\cdot|s,a)^\top \alpha \right| \leqslant 2\sqrt{\frac{\mathsf{supp}\left(\widehat{P}_m(\cdot|s,a)\right) \mathbb{V}\left(\widehat{P}_m(\cdot|s,a), \alpha\right)\iota}{N_m^k(s,a)}} + \frac{5S\iota}{N_m^k(s,a)}.$$

*Similarly, conditioned on $\Omega_2$, we have that,*

$$\left| \left( \widehat{P}_m^k - P_m \right)(\cdot|s,a)^\top \alpha \right| \leqslant \sqrt{\frac{2\mathsf{supp}\left(P_m(\cdot|s,a)\right) \mathbb{V}\left(P_m(\cdot|s,a), \alpha\right)\iota}{N_m^k(s,a)}} + \frac{S\iota}{N_m^k(s,a)}.$$

**Trigger property.** Let $\mathcal{K}$ be the set of indexes of episodes in which no update is triggered. By the update rule, it is obvious that $\left|\mathcal{K}^C\right| \leqslant MSA(\log_2(HK)+1) \leqslant MSA\iota$. Let $t_0(k)$ be the first time an update is triggered in the $k$-th episode if there is an update in this episode and otherwise $H+1$. Define $\mathcal{X}_0 = \{(k,t_0(k)) : k \in \mathcal{K}^C\}$ and $\mathcal{X} = \{(k,t) : k \in \mathcal{K}^C, t_0(k)+1 \leqslant t \leqslant H\}$. We will study quantities multiplied by the trigger indicator $\mathbb{1}[(k,t) \notin \mathcal{X}]$, which we denote using an extra "$\breve{\phantom{x}}$".

We will encounter a special type of summation, so we state it here.

**Lemma 11.** *Let $\{w_t^k \geqslant 0 : (k,t) \in [K] \times [H]\}$ be a group of weights, then*

$$\sum_{k=1}^K \sum_{t=1}^H \frac{\mathbb{1}[(k,t) \notin \mathcal{X}]}{N_{m^k}^k(s_t^k, a_t^k)} \leqslant 3MSA\iota, \quad \sum_{k=1}^K \sum_{t=1}^H \sqrt{\frac{w_t^k \mathbb{1}[(k,t) \notin \mathcal{X}]}{N_{m^k}^k(s_t^k, a_t^k)}} \leqslant \sqrt{3MSA\iota \sum_{k=1}^K \sum_{t=1}^H w_t^k \mathbb{1}[(k,t) \notin \mathcal{X}]}.$$

### B.2.1 OPTIMISM

As a standard approach, we need to show that both Algorithm 2 and Algorithm 3 have optimism in value functions.

For Algorithm 2, it is straightforward. For each episode $k$, we choose the optimistic transition $\widetilde{P}^k$ with the maximum possible value. Lemma 9 shows that with probability at least $1 - \delta$, $\Omega_1$ holds, hence the true transition $P$ is inside the confidence set $\mathcal{P}^k$ for all $k \in [K]$. Therefore, $\widetilde{V}^k \geqslant V^\star$.

Algorithm 3 relies on an important function introduced by Zhang et al. (2021a), so we cite it here:

**Lemma 12** (Adapted from Lemma 14 in Zhang et al. (2021a)). *For any fixed dimension $D$ and two constants $c_1, c_2$ satisfying $c_1^2 \leqslant c_2$, let $f : \Delta([D]) \times \mathbb{R}^D \times \mathbb{R} \times \mathbb{R} \to \mathbb{R}$ with $f(p,v,n,\iota) = pv + \max\left\{ c_1\sqrt{\frac{\mathbb{V}(p,v)\iota}{n}}, c_2\frac{\iota}{n} \right\}$. Then for all $p \in \Delta([D]), \|v\|_\infty \leqslant 1$ and $n,\iota > 0$,*

1. *$f(p,v,n,\iota)$ is non-decreasing in $v$, i.e.,*

   *$\forall v, v'$ such that $\|v\|_\infty, \|v'\|_\infty \leqslant 1, v \leqslant v'$, it holds that $f(p,v,n,\iota) \leqslant f(p,v',n,\iota)$;*

2. $f(p, v, n, \iota) \geqslant pv + \frac{c_1}{2}\sqrt{\frac{\mathbb{V}(p,v)\iota}{n}} + \frac{c_2}{2}\frac{\iota}{n}$.

Due to the complex structure of LMDP, we cannot prove the strong optimism in Zhang et al. (2021a). This is because in LMDP, the optimal policy cannot maximize *all* alpha vectors simultaneously, hence the optimal alpha vectors are not unique. As Algorithm 2, we can only show the optimism at the first step, which is stated in Lemma 13.

**Lemma 13** (Optimism of Algorithm 3). *Algorithm 3 satisfies that: Conditioned on $\Omega_1$, for any episode $k \in [K]$, $\widetilde{V}^k \geqslant V^\star$.*

### B.2.2 REGRET DECOMPOSITION

We introduce the Bellman error here. It contributes to the main order term in the regret.

**Lemma 14** (Bellman error). *Both Algorithm 2 and Algorithm 3 satisfy the following Bellman error bound: Conditioned on $\Omega_1$ and $\Omega_2$, for any $(m, h, a, k) \in [M] \times \mathcal{H} \times \mathcal{A} \times [K]$,*

$$\underbrace{\widetilde{\alpha}_m^k(h, a) - R_m(s, a) - P_m(\cdot|s, a)^\top \widetilde{\alpha}_m^k(har\cdot)}_{\textcircled{1}} \leqslant \min\{\beta_m^k(h, a),\, 1\}, \tag{8}$$

*where $r = R_m(s, a)$ and*

$$\beta_m^k(h, a) = 7\sqrt{\frac{\Gamma\mathbb{V}\left(P_m(\cdot|s, a), \widetilde{\alpha}_m^k(har\cdot)\right)\iota}{N_m^k(s, a)}} + \frac{30S\iota}{N_m^k(s, a)}.$$

Throughout the proof, we denote $\breve{\beta}_t^k = \beta_{m^k}^k(h_t^k, a_t^k)\mathbb{1}[(k, t) \notin \mathcal{X}]$.

Assume that optimism holds, then it is more natural to bound $\widetilde{V}^k - V^{\pi^k}$ instead of $V^\star - V^{\pi^k}$, because the underlying policies are the same for the former case. With simple manipulation, we decompose the regret into $X_1$ the Monte Carlo estimation term for the optimistic value, $X_2$ the Monte Carlo estimation term for the true value, $X_3$ the model estimation error, $X_4$ the Bellman error (*main order term*), and $\left|\mathcal{K}^C\right|$ the correction term for $\mathbb{1}[(k, t) \notin \mathcal{X}]$.

$$\mathbf{Regret}(K) = \sum_{k=1}^K \left(V^\star - V^{\pi^k}\right) \leqslant \sum_{k=1}^K \left(\widetilde{V}^k - V^{\pi^k}\right)$$

$$= \underbrace{\sum_{k=1}^K \left(\widetilde{V}^k - \widetilde{\alpha}_{m^k}^k(s_1^k)\right)}_{X_1} + \sum_{k=1}^K \left(\widetilde{\alpha}_{m^k}^k(s_1^k) - \sum_{t=1}^H \breve{r}_t^k\right) + \underbrace{\sum_{k=1}^K \left(\sum_{t=1}^H \breve{r}_t^k - V^{\pi^k}\right)}_{X_2}$$

$$\overset{(i)}{=} X_1 + X_2 + \sum_{k=1}^K \sum_{t=1}^H \underbrace{\left(\breve{\alpha}_{m^k}^k(h_t^k) - \breve{r}_t^k - P_{m^k}(\cdot|s_t^k, a_t^k)^\top \breve{\alpha}_{m^k}^k(h_t^k a_t^k r_t^k \cdot)\mathbb{1}[(k, t) \notin \mathcal{X}]\right)}_{\leqslant \breve{\beta}_t^k}$$

$$+ \underbrace{\sum_{k=1}^K \sum_{t=1}^H \left(P_{m^k}(\cdot|s_t^k, a_t^k)^\top \breve{\alpha}_{m^k}^k(h_t^k a_t^k r_t^k \cdot) - \breve{\alpha}_{m^k}^k(h_{t+1}^k)\right)}_{X_3}$$

$$+ \sum_{k=1}^K \sum_{t=1}^H P_{m^k}(\cdot|s_t^k, a_t^k)^\top \breve{\alpha}_{m^k}^k(h_t^k a_t^k r_t^k \cdot)(\mathbb{1}[(k, t) \notin \mathcal{X}] - \mathbb{1}[(k, t+1) \notin \mathcal{X}])$$

$$\overset{(ii)}{\leqslant} X_1 + X_2 + X_3 + \underbrace{\sum_{k=1}^K \sum_{t=1}^H \breve{\beta}_t^k}_{X_4} + \sum_{k, t = t_0(k)} \widetilde{P}_{m^k}^k(\cdot|s_t^k, a_t^k)^\top \widetilde{\alpha}_{m^k}^k(h_t^k a_t^k r_t^k \cdot)$$

$$\overset{(iii)}{\leqslant} X_1 + X_2 + X_3 + X_4 + \left|\mathcal{K}^C\right|,$$

where (i) is by $(k, 1) \in \mathcal{X}$ so $\widetilde{\alpha}_{m^k}^k(s_1^k) = \breve{\alpha}_{m^k}^k(s_1^k)$; (ii) follows by Lemma 14 and checking the difference between $\mathbb{1}[(k, t) \notin \mathcal{X}]$ and $\mathbb{1}[(k, t+1) \notin \mathcal{X}]$; (iii) is from the fact that $\widetilde{\alpha}^k \leqslant 1$, and the definition of $t_0(k)$ and $\mathcal{K}$.

### B.2.3 BOUNDING EACH TERM

We start from the easier terms $X_1$ and $X_2$.

**Lemma 15.** *With probability at least $1 - \delta$, we have that $X_1 \leqslant \sqrt{K\iota}$.*

**Lemma 16.** *With probability at least $1 - \delta$, we have that $X_2 \leqslant \sqrt{K\iota}$.*

$X_3$ is a martingale difference sequence. However, if we want to avoid polynomial dependency of $H$, we cannot apply the Azuma's inequality which scales as $\sqrt{H}$. Instead, we use a variance-dependent martingale bound, and this changes $X_3$ into a lower-order term of $X_4$.

**Lemma 17.** *With probability at least $1 - \delta$, we have that $X_3 \leqslant 2\sqrt{2X_4\iota} + 5\iota$.*

Here we show the bound for $X_4$ and its proof first, next we prove Theorem 1. When bounding $X_4$, we are faced with another quantity $X_5$, which is the summation of variances. We do not bound $X_5$ explicitly. Instead, we derive a relation between $X_5$ and $X_4$ (Lemma 19), so finally we solve an inequality of $X_4$.

**Lemma 18.** *Conditioned on $\Omega_1$ and $\Omega_2$, with probability at least $1 - \delta$, we have that $X_4 \leqslant 46\sqrt{MS^2AK\iota} + 947MS^2A\iota^2$.*

*Proof.* From Lemma 14 and Lemma 11, we have that

$$X_4 \leqslant 13\sqrt{M\Gamma SA\iota^2 \underbrace{\sum_{k=1}^{K}\sum_{t=1}^{H} \mathbb{V}\left(P_{m^k}(\cdot|s_t^k, a_t^k), \widetilde{\alpha}_{m^k}^k(h_t^k a_t^k r_t^k \cdot)\right) \mathbb{1}[(k,t) \notin \mathcal{X}]}_{X_5}} + 90MS^2A\iota^2.$$

Applying Lemma 19, using $\sqrt{x+y} \leqslant \sqrt{x} + \sqrt{y}$, and loosening the constants, we have the following inequality:

$$X_4 \leqslant 23\sqrt{M\Gamma SAK\iota^2} + 209MS^2A\iota^2 + 23\sqrt{M\Gamma SA\iota^2} \cdot \sqrt{X_4}.$$

Since $x \leqslant a + b\sqrt{x}$ implies $x \leqslant b^2 + 2a$, we finally have

$$X_4 \leqslant 46\sqrt{M\Gamma SAK\iota} + 947MS^2A\iota^2.$$

This completes the proof. $\qquad\square$

We use the technique of higher-order variance expansion used by (Zhang et al., 2021a) to draw the relation between $X_5$ and $X_4$.

**Lemma 19.** *Conditioned on $\Omega_1$ and $\Omega_2$, with probability at least $1 - \delta$, we have that $X_5 \leqslant 3(K + X_4) + 83\iota$.*

### B.2.4 PROOF OF THEOREM 1

Finally, we are able to prove the main theorem.

*Proof.* From Lemma 15, Lemma 16, Lemma 17 and property of $\mathcal{K}$, we have that, with probability at least $1 - 3\delta$,

$$\mathbf{Regret}(K) \leqslant 2\sqrt{K\iota} + 2\sqrt{2X_4\iota} + 5\iota + X_4 + MSA\iota.$$

Plugging in Lemma 18, using $\sqrt{x+y} \leqslant \sqrt{x} + \sqrt{y}$, we finally have

$$\mathbf{Regret}(K) \leqslant 68\sqrt{MS^2AK\iota} + 1041MS^2A\iota^2$$

holds with probability at least $1 - 6\delta$ (using Lemma 9). Rescaling $\delta \leftarrow \frac{\delta}{6}$ completes the proof. $\quad\square$

**Lemma 9.** $\mathbb{P}[\Omega_1], \mathbb{P}[\Omega_2] \geqslant 1 - \delta$.

*Proof.* From Lemma 5 we have that, for any fixed $(m, s, a, s', k) \in [M] \times \mathcal{S} \times \mathcal{A} \times \mathcal{S} \times [K]$ and $2 \leqslant N_m^k(s, a) \leqslant HK$,

$$\mathbb{P}\left[\left|\left(\widehat{P}_m^k - P_m\right)(s'|s,a)\right| > \sqrt{\frac{2\widehat{P}_m^k(s'|s,a)\iota'}{N_m^k(s,a) - 1}} + \frac{7\iota'}{3(N_m^k(s,a) - 1)}\right] \leqslant \frac{\delta}{M \cdot S \cdot A \cdot S \cdot K \cdot HK},$$

where $\iota' = \ln\left(\frac{2MS^2AHK^2}{\delta}\right) \leqslant \iota$. From $\frac{1}{x-1} \leqslant \frac{2}{x}$ when $x \geqslant 2$ ($N_m^k(s,a) = 1$ is trivial), and applying union bound over all possible events, we have that $\mathbb{P}[\cap_{k=1}^K \Omega_1^k] \geqslant 1 - \delta$.

From Lemma 4 we have that, for any fixed $(m, s, a, s', k) \in [M] \times \mathcal{S} \times \mathcal{A} \times \mathcal{S} \times [K]$ and $1 \leqslant N_m^k(s,a) \leqslant HK$,

$$\mathbb{P}\left[\left|(\widehat{P}_m - P_m)(s'|s,a)\right| > \sqrt{\frac{2P_m(s'|s,a)\iota'}{N_m^k(s,a)}} + \frac{\iota'}{N_m^k(s,a)}\right] \leqslant \frac{\delta}{M \cdot S \cdot A \cdot S \cdot K \cdot HK}.$$

By taking a union bound over all possible events, we have that $\mathbb{P}[\cap_{k=1}^K \Omega_2^k] \geqslant 1 - \delta$. $\qquad\square$

**Lemma 10.** *Conditioned on $\Omega_1$, we have that for any $(m, s, a, k) \in [M] \times \mathcal{S} \times \mathcal{A} \times [K]$, and any $S$-dimensional vector $\alpha$ such that $\|\alpha\|_\infty \leqslant 1$,*

$$\left|\left(\widehat{P}_m^k - P_m\right)(\cdot|s,a)^\top \alpha\right| \leqslant 2\sqrt{\frac{\text{supp}\left(\widehat{P}_m(\cdot|s,a)\right)\mathbb{V}\left(\widehat{P}_m(\cdot|s,a),\alpha\right)\iota}{N_m^k(s,a)}} + \frac{5S\iota}{N_m^k(s,a)}.$$

*Similarly, conditioned on $\Omega_2$, we have that,*

$$\left|\left(\widehat{P}_m^k - P_m\right)(\cdot|s,a)^\top \alpha\right| \leqslant \sqrt{\frac{2\text{supp}\left(P_m(\cdot|s,a)\right)\mathbb{V}\left(P_m(\cdot|s,a),\alpha\right)\iota}{N_m^k(s,a)}} + \frac{S\iota}{N_m^k(s,a)}.$$

*Proof.* We fix the episode number $k$ and omit it for simplicity.

$$\left|\left(\widehat{P}_m - P_m\right)(\cdot|s,a)^\top \alpha\right|$$

$$\overset{(i)}{=} \left|\sum_{s' \in \mathcal{S}} \left(\widehat{P}_m - P_m\right)(s'|s,a)\left(\alpha(s') - \widehat{P}_m(\cdot|s,a)^\top \alpha\right)\right|$$

$$\leqslant \sum_{s' \in \mathcal{S}} \left|\widehat{P}_m - P_m\right|(s'|s,a)\left|\alpha(s') - \widehat{P}_m(\cdot|s,a)^\top \alpha\right|$$

$$\overset{(ii)}{\leqslant} \sum_{s' \in \mathcal{S}} \left(2\sqrt{\frac{\widehat{P}_m(s'|s,a)\iota}{N_m(s,a)}} + \frac{5\iota}{N_m(s,a)}\right)\left|\alpha(s') - \widehat{P}_m(\cdot|s,a)^\top \alpha\right|$$

$$\leqslant 2\sqrt{\frac{\iota}{N_m(s,a)}} \sum_{s' \in \mathcal{S}} \mathbb{1}\left[\widehat{P}_m(s'|s,a) > 0\right]\sqrt{\widehat{P}_m(s'|s,a)}\left|\alpha(s') - \widehat{P}_m(\cdot|s,a)^\top \alpha\right| + \frac{5S\iota}{N_m(s,a)}$$

$$\overset{(iii)}{\leqslant} 2\sqrt{\frac{\iota}{N_m(s,a)}}\sqrt{\sum_{s' \in \mathcal{S}} \mathbb{1}\left[\widehat{P}_m(s'|s,a) > 0\right] \cdot \sum_{s' \in \mathcal{S}} \widehat{P}_m(s'|s,a)\left(\alpha(s') - \widehat{P}_m(\cdot|s,a)^\top \alpha\right)^2} + \frac{5S\iota}{N_m(s,a)}$$

$$= 2\sqrt{\frac{\text{supp}\left(\widehat{P}_m(\cdot|s,a)\right)\mathbb{V}\left(\widehat{P}_m(\cdot|s,a),\alpha\right)\iota}{N_m(s,a)}} + \frac{5S\iota}{N_m(s,a)},$$

where (i) comes from that $P_m(\cdot|s,a)^\top \alpha$ is a constant and $\widehat{P}, P$ are two distributions; (ii) is by the definition of $\Omega_1^k$ (Equation (6)) and $\|\alpha\|_\infty \leqslant 1$; (iii) is from the Cauchy-Schwarz inequality. The second part is similar. $\qquad\square$

**Lemma 11.** *Let $\{w_t^k \geqslant 0 : (k,t) \in [K] \times [H]\}$ be a group of weights, then*

$$\sum_{k=1}^{K} \sum_{t=1}^{H} \frac{\mathbb{1}[(k,t) \notin \mathcal{X}]}{N_{m^k}^k(s_t^k, a_t^k)} \leqslant 3MSA\iota, \quad \sum_{k=1}^{K} \sum_{t=1}^{H} \sqrt{\frac{w_t^k \mathbb{1}[(k,t) \notin \mathcal{X}]}{N_{m^k}^k(s_t^k, a_t^k)}} \leqslant \sqrt{3MSA\iota \sum_{k=1}^{K} \sum_{t=1}^{H} w_t^k \mathbb{1}[(k,t) \notin \mathcal{X}]}.$$

*Proof.* From Algorithm 1 and the definition of $\mathcal{K}$, we have that for any $i \in \mathbb{N}, (m,s,a) \in [M] \times \mathcal{S} \times \mathcal{A}$,

$$\sum_{k=1}^{K} \sum_{t=1}^{H} \mathbb{1}[(m^k, s_t^k, a_t^k) = (m,s,a), N_m^k(s,a) = 2^i, (k,t) \notin \mathcal{X}] \leqslant \left\{ \begin{array}{ll} 2, & i = 0, \\ 2^i, & i \geqslant 1. \end{array} \right.$$

So

$$\sum_{k=1}^{K} \sum_{t=1}^{H} \frac{\mathbb{1}[(k,t) \notin \mathcal{X}]}{N_{m^k}^k(s_t^k, a_t^k)}$$

$$= \sum_{(m,s,a) \in [M] \times \mathcal{S} \times \mathcal{A}} \sum_{i=0}^{\lfloor \log_2(HK) \rfloor} \sum_{k=1}^{K} \sum_{t=1}^{H} \mathbb{1}[(m^k, s_t^k, a_t^k) = (m,s,a), N_m^k(s,a) = 2^i] \frac{\mathbb{1}[(k,t) \notin \mathcal{X}]}{2^i}$$

$$\leqslant \sum_{(m,s,a) \in [M] \times \mathcal{S} \times \mathcal{A}} \left( 2 + \sum_{i=1}^{\lfloor \log_2(HK) \rfloor} 1 \right)$$

$$\leqslant 3MSA\iota.$$

Therefore,

$$\sum_{k=1}^{K} \sum_{t=1}^{H} \sqrt{\frac{w_t^k \mathbb{1}[(k,t) \notin \mathcal{X}]}{N_{m^k}^k(s_t^k, a_t^k)}} \stackrel{(i)}{=} \sum_{k=1}^{K} \sum_{t=1}^{H} \sqrt{w_t^k \mathbb{1}[(k,t) \notin \mathcal{X}] \cdot \frac{\mathbb{1}[(k,t) \notin \mathcal{X}]}{N_{m^k}^k(s_t^k, a_t^k)}}$$

$$\stackrel{(ii)}{\leqslant} \sqrt{\left( \sum_{k=1}^{K} \sum_{t=1}^{H} w_t^k \mathbb{1}[(k,t) \notin \mathcal{X}] \right) \left( \sum_{k=1}^{K} \sum_{t=1}^{H} \frac{\mathbb{1}[(k,t) \notin \mathcal{X}]}{N_{m^k}^k(s_t^k, a_t^k)} \right)}$$

$$\leqslant \sqrt{3MSA\iota \sum_{k=1}^{K} \sum_{t=1}^{H} w_t^k \mathbb{1}[(k,t) \notin \mathcal{X}]},$$

where (i) is by the property of indicator function; (ii) is by the Cauchy-Schwarz inequality. $\square$

**Lemma 13** (Optimism of Algorithm 3). *Algorithm 3 satisfies that: Conditioned on $\Omega_1$, for any episode $k \in [K]$, $\widetilde{V}^k \geqslant V^\star$.*

*Proof.* We first argue that for any policy $\pi$ and any episode $k$, we have that $\widetilde{V}_{\mathcal{M}^k}^\pi \geqslant V^\pi$. Throughout the proof, the episode number $k$ is fixed and omitted in any superscript. We proceed the proof for $h$ in the order $\mathcal{H}_H, \mathcal{H}_{H-1}, \ldots, \mathcal{H}_1$, using induction. Recall that for any $h \in \mathcal{H}_{H+1}$ we define $\widetilde{\alpha}_m^\pi(h, a) = \alpha_m^\pi(h, a) = 0$. Now suppose for time step $t$, we already have $\widetilde{\alpha}_m^\pi(h', a) \geqslant \alpha_m^\pi(h', a)$ for any $h' \in \mathcal{H}_{t+1}$, then $\widetilde{\alpha}_m^\pi(h') = \widetilde{\alpha}_m^\pi(h', \pi(h')) \geqslant \alpha_m^\pi(h', \pi(h')) = \alpha_m^\pi(h')$. For any $h \in \mathcal{H}_t$,

$$\widetilde{\alpha}_m^\pi(h, a)$$

$$\stackrel{(i)}{=} \min \left\{ R_m(s,a) + B_m(h,a) + \widehat{P}_m(\cdot|s,a)^\top \widetilde{\alpha}_m^\pi(har\cdot), 1 \right\}$$

$$= \min \left\{ R_m(s,a) + \max \left\{ 4\sqrt{\frac{\mathsf{supp}\left(\widehat{P}_m(\cdot|s,a)\right) \mathbb{V}\left(\widehat{P}_m(\cdot|s,a), \widetilde{\alpha}_m^\pi(har\cdot)\right) \iota}{N_m(s,a)}}, \frac{16S\iota}{N_m(s,a)} \right\} \right.$$

$$\left. + \widehat{P}_m(\cdot|s,a)^\top \widetilde{\alpha}_m^\pi(har\cdot), 1 \right\}$$

$$\overset{\text{(ii)}}{=} \min \left\{ R_m(s,a) + f\left( \widehat{P}_m(\cdot|s,a), \widetilde{\alpha}_m^\pi(har\cdot), N_m(s,a), \iota \right), 1 \right\}$$

$$\overset{\text{(iii)}}{\geqslant} \min \left\{ R_m(s,a) + f\left( \widehat{P}_m(\cdot|s,a), \alpha_m^\pi(har\cdot), N_m(s,a), \iota \right), 1 \right\}$$

$$\overset{\text{(iv)}}{\geqslant} \min \left\{ R_m(s,a) + \widehat{P}_m(\cdot|s,a)^\top \alpha_m^\pi(har\cdot) \right.$$
$$\left. + 2\sqrt{\frac{\mathsf{supp}\left( \widehat{P}_m(\cdot|s,a) \right) \mathbb{V}\left( \widehat{P}_m(\cdot|s,a), \alpha_m^\pi(har\cdot) \right) \iota}{N_m(s,a)}} + \frac{8S\iota}{N_m(s,a)}, 1 \right\}$$

$$\overset{\text{(v)}}{\geqslant} \min \left\{ R_m(s,a) + P_m(\cdot|s,a)^\top \alpha_m^\pi(har\cdot), 1 \right\}$$
$$= \alpha_m^\pi(h,a),$$

where (i) is by taking $r = R_m(s,a)$; (ii) is by recognizing $c_1 = 4\sqrt{\mathsf{supp}\left( \widehat{P}_m(\cdot|s,a) \right)}, c_2 = 16S$ in Lemma 12, which satisfy $c_1^2 \leqslant c_2$; (iii) and (iv) come by successively applying the first and second property in Lemma 12; (v) is an implication of Lemma 10, conditioning on $\Omega_1$ and taking $\alpha = \alpha_m^\pi(har\cdot)$.

The proof is completed by the fact that $\pi^k = \arg\max_{\pi\in\Pi} \widetilde{V}^\pi$, hence $\widetilde{V}^k \geqslant \widetilde{V}^{\pi^\star} \geqslant V^\star$. $\qquad\square$

**Lemma 14** (Bellman error). *Both Algorithm 2 and Algorithm 3 satisfy the following Bellman error bound: Conditioned on $\Omega_1$ and $\Omega_2$, for any $(m,h,a,k) \in [M] \times \mathcal{H} \times \mathcal{A} \times [K]$,*

$$\underbrace{\widetilde{\alpha}_m^k(h,a) - R_m(s,a) - P_m(\cdot|s,a)^\top \widetilde{\alpha}_m^k(har\cdot)}_{\text{①}} \leqslant \min\{\beta_m^k(h,a), 1\}, \tag{8}$$

*where $r = R_m(s,a)$ and*

$$\beta_m^k(h,a) = 7\sqrt{\frac{\Gamma \mathbb{V}\left( P_m(\cdot|s,a), \widetilde{\alpha}_m^k(har\cdot) \right) \iota}{N_m^k(s,a)} + \frac{30S\iota}{N_m^k(s,a)}}.$$

*Proof.* Here we decompose the Bellman error in a generic way. We use $\widetilde{P}$ for the transition and $B$ for the bonus used in the optimistic model. For Algorithm 2, $B = 0$; while for Algorithm 3, $\widetilde{P} = \widehat{P}$.

The upper bound of 1 is trivial. Fix any $(m,h,a,k) \in [M] \times \mathcal{H} \times \mathcal{A} \times [K]$, then

$$\text{①} = R_m(s,a) + B_m^k(h,a) + \widetilde{P}_m^k(\cdot|s,a)^\top \widetilde{\alpha}_m^k(har\cdot) - R_m(s,a) - P_m(\cdot|s,a)^\top \widetilde{\alpha}_m^k(har\cdot)$$
$$= B_m^k(h,a) + \left( \widetilde{P}_m^k - P_m \right)(\cdot|s,a)^\top \widetilde{\alpha}_m^k(har\cdot).$$

Next we proceed in two ways.

For Algorithm 2, we utilize Lemma 20 and a similar argument as Lemma 10. It gives

$$\text{①}_{\text{Bernstein}} \leqslant 4\sqrt{\frac{\mathsf{supp}\left( P_m(\cdot|s,a) \right) \mathbb{V}\left( P_m(\cdot|s,a), \alpha \right) \iota}{N_m^k(s,a)}} + \frac{30S\iota}{N_m^k(s,a)}.$$

For Algorithm 3, we plug in the definition of $B$ and use Lemma 10. It gives

$$\text{①}_{\text{MVP}} \leqslant 4\sqrt{\frac{\mathsf{supp}\left( \widehat{P}_m^k(\cdot|s,a) \right) \mathbb{V}\left( \widehat{P}_m^k(\cdot|s,a), \widetilde{\alpha}_m^k(har\cdot) \right) \iota}{N_m^k(s,a)}}$$
$$+ \sqrt{\frac{2\mathsf{supp}\left( P_m(\cdot|s,a) \right) \mathbb{V}\left( P_m(\cdot|s,a), \widetilde{\alpha}_m^k(har\cdot) \right) \iota}{N_m^k(s,a)}} + \frac{17S\iota}{N_m^k(s,a)}.$$

Next we bound $\mathbb{V}\left(\widehat{P}_m^k(\cdot|h,a), \widetilde{\alpha}_m^k(har\cdot)\right)$.

$$
\mathbb{V}\left(\widehat{P}_m^k(\cdot|h,a), \widetilde{\alpha}_m^k(har\cdot)\right)
$$

$$
= \sum_{s'\in\mathcal{S}} \widehat{P}_m^k(s'|s,a)\left(\widetilde{\alpha}_m^k(hars') - \widehat{P}_m^k(\cdot|s,a)^\top\widetilde{\alpha}_m^k(har\cdot)\right)^2
$$

$$
\overset{(i)}{\leqslant} \sum_{s'\in\mathcal{S}} \widehat{P}_m^k(s'|s,a)\left(\widetilde{\alpha}_m^k(hars') - P_m(\cdot|s,a)^\top\widetilde{\alpha}_m^k(har\cdot)\right)^2
$$

$$
\overset{(ii)}{\leqslant} \sum_{s'\in\mathcal{S}} \left(P_m(s'|s,a) + \sqrt{\frac{2P_m(s'|s,a)\iota}{N_m^k(s,a)}} + \frac{\iota}{N_m^k(s,a)}\right)\left(\widetilde{\alpha}_m^k(hars') - P_m(\cdot|s,a)^\top\widetilde{\alpha}_m^k(har\cdot)\right)^2
$$

$$
\leqslant \sum_{s'\in\mathcal{S}} \left(\frac{3}{2}P_m(s'|s,a) + \frac{2\iota}{N_m^k(s,a)}\right)\left(\widetilde{\alpha}_m^k(hars') - P_m(\cdot|s,a)^\top\widetilde{\alpha}_m^k(har\cdot)\right)^2
$$

$$
\leqslant \frac{3}{2}\mathbb{V}\left(P_m(\cdot|s,a), \widetilde{\alpha}_m^k(har\cdot)\right) + \frac{2S\iota}{N_m^k(s,a)},
$$

where (i) is by that $z^\star = \sum_i p_i x_i$ minimizes $f(z) = \sum_i p_i(x_i - z)^2$; (ii) is by $\Omega_2$. Finally, using $\sqrt{x+y} \leqslant \sqrt{x} + \sqrt{y}$ and $\mathsf{supp}\left(\widehat{P}_m^k(\cdot|s,a)\right) \leqslant \mathsf{supp}\left(P_m(\cdot|s,a)\right)$, we have

$$
\textcircled{1}_{\mathrm{MVP}} \leqslant 7\sqrt{\frac{\mathsf{supp}\left(P_m(\cdot|s,a)\right)\mathbb{V}\left(P_m(\cdot|s,a), \widetilde{\alpha}_m^k(har\cdot)\right)\iota}{N_m^k(s,a)}} + \frac{23S\iota}{N_m^k(s,a)}.
$$

Therefore, setting

$$
\beta_m^k(h,a) = 7\sqrt{\frac{\Gamma\mathbb{V}\left(P_m(\cdot|s,a), \widetilde{\alpha}_m^k(har\cdot)\right)\iota}{N_m^k(s,a)}} + \frac{30S\iota}{N_m^k(s,a)}
$$

completes the proof. $\qquad\square$

**Lemma 15.** *With probability at least $1 - \delta$, we have that $X_1 \leqslant \sqrt{K\iota}$.*

*Proof.* By definition, $V^k = \sum_{m=1}^M \sum_{s\in\mathcal{S}} w_m \nu_m(s)\alpha_m^k(s)$. Thus, $\alpha_{m^k}^k(s_1^k)$ is a random variable with mean $V^k$. Also, $\alpha_{m^k}^k(s_1^k)$ is measurable with respect to $\bar{U}^{k-1}$. Using Lemma 3 and $\left|\alpha_{m^k}^k(s_1^k) - V^k\right| \leqslant 1$, we have

$$
\mathbb{P}\left[X_1 > \sqrt{K\iota}\right] \leqslant \delta.
$$

This completes the proof. $\qquad\square$

**Lemma 16.** *With probability at least $1 - \delta$, we have that $X_2 \leqslant \sqrt{K\iota}$.*

*Proof.* By definition, $X_2 \leqslant \sum_{k=1}^K \left(\sum_{t=1}^H r_t^k - V^{\pi^k}\right)$. From Monte Carlo simulation, $\mathbb{E}\left[\sum_{t=1}^H r_t^k\right] = V^{\pi^k}$. Also, $\sum_{t=1}^H r_t^k$ is measurable with respect to $\bar{U}^{k-1}$. Using Lemma 3 and $\left|\sum_{t=1}^H r_t^k - V^{\pi^k}\right| \leqslant 1$, we have

$$
\mathbb{P}\left[\sum_{k=1}^K \left(\sum_{t=1}^H r_t^k - V^{\pi^k}\right) > \sqrt{K\iota}\right] \leqslant \delta.
$$

This completes the proof. $\qquad\square$

**Lemma 17.** *With probability at least $1 - \delta$, we have that $X_3 \leqslant 2\sqrt{2X_4\iota} + 5\iota$.*

*Proof.* Observe that $\mathbb{1}[(k, t+1) \notin \mathcal{X}] \leqslant \mathbb{1}[(k,t) \notin \mathcal{X}]$, so

$$X_3 \leqslant \sum_{k=1}^{K} \sum_{t=1}^{H} \left( P_{m^k}(\cdot | s_t^k, a_t^k)^\top \widetilde{\alpha}_{m^k}^k(h_t^k a_t^k r_t^k \cdot) - \widetilde{\alpha}_{m^k}^k(h_{t+1}^k) \right) \mathbb{1}[(k,t) \notin \mathcal{X}].$$

This is a martingale. By taking $c = 1$ in Lemma 6, we have

$$\mathbb{P}\left[ X_3 > 2\sqrt{2X_4 \iota} + 5\iota \right] \leqslant \delta.$$

This completes the proof. $\qquad\qquad\square$

**Lemma 19.** *Conditioned on $\Omega_1$ and $\Omega_2$, with probability at least $1 - \delta$, we have that $X_5 \leqslant 3(K + X_4) + 83\iota$.*

*Proof.* For any non-negative integer $d$, define

$$F(d) := \sum_{k=1}^{K} \sum_{t=1}^{H} \left( P_{m^k}(\cdot | s_t^k, a_t^k)^\top \left( \widetilde{\alpha}_{m^k}^k(h_t^k a_t^k r_t^k \cdot) \right)^{2^d} - \left( \widetilde{\alpha}_{m^k}^k(h_{t+1}^k) \right)^{2^d} \right) \mathbb{1}[(k,t) \notin \mathcal{X}],$$

$$G(d) := \sum_{k=1}^{K} \sum_{t=1}^{H} \mathbb{V}\left( P_{m^k}(\cdot | s_t^k, a_t^k), \left( \widetilde{\alpha}_{m^k}^k(h_t^k a_t^k r_t^k \cdot) \right)^{2^d} \right) \mathbb{1}[(k,t) \notin \mathcal{X}].$$

Then $X_5 = G(0)$. Direct computation gives that

$G(d)$

$$= \sum_{k=1}^{K} \sum_{t=1}^{H} \left( P_{m^k}(\cdot | s_t^k, a_t^k) \left( \widetilde{\alpha}_{m^k}^k(h_t^k a_t^k r_t^k \cdot) \right)^{2^{d+1}} - \left( P_{m^k}(\cdot | s_t^k, a_t^k) \left( \widetilde{\alpha}_{m^k}^k(h_t^k a_t^k r_t^k \cdot) \right)^{2^d} \right)^2 \right) \mathbb{1}[(k,t) \notin \mathcal{X}]$$

$$\overset{(i)}{\leqslant} \sum_{k=1}^{K} \sum_{t=1}^{H} \left( P_{m^k}(\cdot | s_t^k, a_t^k) \left( \widetilde{\alpha}_{m^k}^k(h_t^k a_t^k r_t^k \cdot) \right)^{2^{d+1}} - \left( \widetilde{\alpha}_{m^k}^k(h_{t+1}^k) \right)^{2^{d+1}} \right) \mathbb{1}[(k,t) \notin \mathcal{X}] + \underbrace{\left( \widetilde{\alpha}_{m^k}^k(h_{H+1}^k) \right)^{2^{d+1}}}_{=0}$$

$$+ \sum_{k=1}^{K} \sum_{t=1}^{H} \left( \left( \widetilde{\alpha}_{m^k}^k(h_t^k) \right)^{2^{d+1}} - \left( P_{m^k}(\cdot | s_t^k, a_t^k) \widetilde{\alpha}_{m^k}^k(h_t^k a_t^k r_t^k \cdot) \right)^{2^{d+1}} \right) \mathbb{1}[(k,t) \notin \mathcal{X}] \underbrace{- \left( \widetilde{\alpha}_{m^k}^k(s_1^k) \right)^{2^{d+1}}}_{\leqslant 0}$$

$$\overset{(ii)}{\leqslant} F(d+1) + 2^{d+1} \sum_{k=1}^{K} \sum_{t=1}^{H} \max\left\{ \widetilde{\alpha}_{m^k}^k(h_t^k, a_t^k) - P_{m^k}(\cdot | s_t^k, a_t^k)^\top \widetilde{\alpha}_{m^k}^k(h_t^k a_t^k r_t^k \cdot), \, 0 \right\} \mathbb{1}[(k,t) \notin \mathcal{X}]$$

$$= F(d+1) + 2^{d+1} \sum_{k=1}^{K} \sum_{t=1}^{H} \max\left\{ \widetilde{\alpha}_{m^k}^k(h_t^k) - P_{m^k}(\cdot | s_t^k, a_t^k)^\top \widetilde{\alpha}_{m^k}^k(h_t^k a_t^k r_t^k \cdot), \, 0 \right\} \mathbb{1}[(k,t) \notin \mathcal{X}]$$

$$\overset{(iii)}{\leqslant} F(d+1) + 2^{d+1} \sum_{k=1}^{K} \sum_{t=1}^{H} (\breve{r}_t^k + \breve{\beta}_t^k)$$

$$\overset{(iv)}{\leqslant} F(d+1) + 2^{d+1}(K + X_4),$$

where (i) is by convexity of function $x^{2^d}$; (ii) is by $x^{2^d} - y^{2^d} \leqslant 2^d \max\{x - y, \, 0\}$ for $x, y \in [0, 1]$; (iii) comes from Lemma 14; (iv) is by the assumption that reward within an episode is upper-bounded by 1 and the definiton of $X_4$.

For a fixed $d$, $F(d)$ is a martingale. By taking $c = 1$ in Lemma 6, we have

$$\mathbb{P}\left[ F(d) > 2\sqrt{2G(d)(\log_2(HK) + \ln(2/\delta))} + 5(\log_2(HK) + \ln(2/\delta)) \right] \leqslant \delta.$$

Taking $\delta' = \delta/(\log_2(HK) + 1)$, using $x \geqslant \ln(x) + 1$ and finally swapping $\delta$ and $\delta'$, we have that

$$\mathbb{P}\left[ F(d) > 2\sqrt{2G(d)(2\log_2(HK) + \ln(2/\delta))} + 5(2\log_2(HK) + \ln(2/\delta)) \right] \leqslant \frac{\delta}{\log_2(HK) + 1}.$$

Taking a union bound over $d = 1, 2, \ldots, \log_2(HK)$, we have that with probability at least $1 - \delta$,

$$F(d) \leqslant 4\sqrt{(F(d+1) + 2^{d+1}(K + X_4))\iota} + 10\iota.$$

From Lemma 7, taking $\lambda_1 = HK$, $\lambda_2 = 4\sqrt{\iota}$, $\lambda_3 = K + X_4$, $\lambda_4 = 10\iota$, we have that

$$F(1) \leqslant \max\{(4\sqrt{\iota} + \sqrt{26\iota})^2,\ 8\sqrt{2(K + X_4)\iota} + 10\iota\} \overset{\text{(i)}}{\leqslant} K + X_4 + 83\iota,$$

where (i) uses $\sqrt{xy} \leqslant \frac{x+y}{2}$ and $\max\{x, y\} \leqslant x + y$ for $x, y \geqslant 0$. Hence

$$X_5 = G(0) \leqslant F(1) + 2(K + X_4) \leqslant 3(K + X_4) + 83\iota.$$

This completes the proof. $\qquad\square$

**Lemma 20.** *Conditioned on $\Omega_1$, we have that for any $(k, m, s, a, s') \in [K] \times [M] \times \mathcal{S} \times \mathcal{A} \times \mathcal{S}$,*

$$\left| P_m(s'|s, a) - \widetilde{P}_m^k(s'|s, a) \right| \leqslant 4\sqrt{\frac{P_m(s'|s, a)\iota}{N_m^k(s, a)}} + \frac{30\iota}{N_m^k(s, a)}.$$

*Proof.* From $\Omega_1$ we have

$$\widehat{P}_m^k(s'|s, a) \leqslant 2\sqrt{\frac{\widehat{P}_m^k(s'|s, a)\iota}{N_m^k(s, a)}} + \frac{5\iota}{N_m^k(s, a)} + P_m(s'|s, a).$$

This is a quadratic inequality in $\sqrt{\widehat{P}_m^k(s'|s, a)}$. Using the fact that $x^2 \leqslant ax + b$ implies $x \leqslant a + \sqrt{b}$ with $a = 2\sqrt{\frac{\iota}{N_m^k(s,a)}}, b = \frac{5\iota}{N_m^k(s,a)} + P_m(s'|s, a)$, and $\sqrt{x + y} \leqslant \sqrt{x} + \sqrt{y}$, we have

$$\sqrt{\widehat{P}_m^k(s'|s, a)} \leqslant \sqrt{P_m(s'|s, a)} + 5\sqrt{\frac{\iota}{N_m^k(s, a)}}.$$

Substituting this into $\Omega$ we have

$$\left| P_m(s'|s, a) - \widehat{P}_m^k(s'|s, a) \right| \leqslant 2\sqrt{\frac{P_m(s'|s, a)\iota}{N_m^k(s, a)}} + \frac{15\iota}{N_m^k(s, a)}.$$

From the construction of $\widetilde{P}_m^k$ we also have

$$\left| \widehat{P}_m^k(s'|s, a) - \widetilde{P}_m^k(s'|s, a) \right| \leqslant 2\sqrt{\frac{P_m(s'|s, a)\iota}{N_m^k(s, a)}} + \frac{15\iota}{N_m^k(s, a)}.$$

Therefore, from triangle inequality we have the desired result. $\qquad\square$

### B.4 PROOF OF THE REGRET LOWER BOUND

**Theorem 2.** *Assume that $S \geqslant 6$, $A \geqslant 2$ and $M \leqslant \lfloor \frac{S}{2} \rfloor!$. For any algorithm $\pi$, there exists an LMDP $\mathcal{M}_\pi$ such that, for $K \geqslant \widetilde{\Omega}(M^2 + MSA)$, its expected regret in $\mathcal{M}_\pi$ after $K$ episodes satisfies*

$$R(\mathcal{M}_\pi, \pi, K) := \mathbb{E}\left[ \sum_{k=1}^{K}(V^\star - V^k) \,\middle|\, \mathcal{M}_\pi, \pi \right] = \Omega\left(\sqrt{MSAK}\right).$$

*Proof.* We need to introduce an alternative regret measure for an *MDP* based on simulating an *LMDP* algorithm. Let $\mathcal{M}(m, \ell^\star, a^\star)$ be an MDP which contains an encoding phase with permutation $\boldsymbol{\sigma}(m)$, and a guessing phase with correct answer $(\ell^\star, a^\star)$. Given any LMDP algorithm $\pi$, a target position $m$ and a pair of LMDP configuration $(\ell^\star, a^\star)$, we can construct an MDP algorithm $\pi(m, \ell^\star, a^\star)$ as in Algorithm 4.

This algorithm admits two types of training: ① When $K$ is specified, it returns after $K$ episodes, regardless of how many times it interacts with the target MDP; ② When $\overline{K}_m$ is specified, it does not return until it interacts with the MDP for $\overline{K}_m$ times, regardless of how many episodes elapse.

**Algorithm 4** $\boldsymbol{\pi}(m, \boldsymbol{\ell}^\star, \boldsymbol{a}^\star)$: an algorithm for an MDP.

---

1: **Input:** an MDP $\mathcal{M}(m, \ell^\star, a^\star)$; an LMDP algorithm $\boldsymbol{\pi}$; a pair of LMDP configuration $(\boldsymbol{\ell}^\star, \boldsymbol{a}^\star)$; specify *exactly one*: a simulation episode budget $K$ or a target interaction episode $\overline{K}_m$.
2: Initialize actual interaction counter $K_m \leftarrow 0$.
3: **for** $k = 1, 2, \ldots$ **do**
4:     Randomly choose $m' \sim \mathsf{Unif}(M)$.
5:     **if** $m^k \neq m$ **then**
6:         Use $\boldsymbol{\pi}$ to interact with the $m^k$th MDP of $\mathcal{M}(\boldsymbol{\ell}^\star, \boldsymbol{a}^\star)$.
7:     **else**
8:         Use $\boldsymbol{\pi}$ to interact with $\mathcal{M}(m, \ell^\star, a^\star)$.
9:         $K_m \leftarrow K_m + 1$.
10:    **end if**
11:    **if** ($K$ is specified and $k = K$) or ($\overline{K}_m$ is specified and $K_m = \overline{K}_m$) **then**
12:        Break.
13:    **end if**
14: **end for**

---

Let $V^\star$ and $V^k$ be the optimal value function and the value function of $\boldsymbol{\pi}(m, \boldsymbol{\ell}^\star, \boldsymbol{a}^\star), K)$ under the MDP $\mathcal{M}(m, \ell^\star, a^\star)$. The alternative regret for MDP (corresponding to ①) is:

$$\widetilde{R}(\mathcal{M}(m, \ell^\star, a^\star), \boldsymbol{\pi}(m, \boldsymbol{\ell}^\star, \boldsymbol{a}^\star), K) := \mathbb{E}\left[ \sum_{k=1}^{K} \mathbb{1}[m^k = m](V^\star - V^k) \ \middle| \ \mathcal{M}(m, \ell^\star, a^\star), \boldsymbol{\pi}(m, \boldsymbol{\ell}^\star, \boldsymbol{a}^\star) \right].$$

Roughly, this is a regret for $K_m$ episodes, though $K_m$ is stochastic.

In our hard instances, the MDPs in the LMDP can be considered separately. So $V^\star = \frac{1}{M} \sum_{m=1}^{M} V_m^\star$, where $V_m^\star$ is *the optimal value function of (which is equal to the value function of the optimal policy applied to)* the $m$th MDP. According to Monte-Carlo sampling,

$$\begin{aligned}
R(\mathcal{M}(\boldsymbol{\ell}^\star, \boldsymbol{a}^\star), \boldsymbol{\pi}, K) &= \mathbb{E}\left[ \sum_{k=1}^{K} (V_{m^k}^\star - V_{m^k}^k) \ \middle| \ \mathcal{M}(\boldsymbol{\ell}^\star, \boldsymbol{a}^\star), \boldsymbol{\pi} \right] \\
&= \sum_{m=1}^{M} \mathbb{E}\left[ \sum_{k=1}^{K} \mathbb{1}[m^k = m](V_{m^k}^\star - V_{m^k}^k) \ \middle| \ \mathcal{M}(\boldsymbol{\ell}^\star, \boldsymbol{a}^\star), \boldsymbol{\pi} \right] \\
&= \sum_{m=1}^{M} \widetilde{R}(\mathcal{M}(m, \ell_m^\star, a_m^\star), \boldsymbol{\pi}(m, \boldsymbol{\ell}^\star, \boldsymbol{a}^\star), K).
\end{aligned}$$

The last step is because the behavior of "focusing on the $m$th MDP in the LMDP" and "using the simulator" are the same. Denote $K_m$ as the number of episodes spent in the $m$th MDP, which is a random variable. According to Lemma 4,

$$\mathbb{P}\left[ \left| \frac{K_m}{K} - \frac{1}{M} \right| > \sqrt{\frac{\frac{2}{M}\left(1 - \frac{1}{M}\right)\ln\left(\frac{2MC^M}{\delta}\right)}{K}} + \frac{\ln\left(\frac{2MC^M}{\delta}\right)}{K} \right] \leq \frac{\delta}{MC^M},$$

which implies

$$\mathbb{P}\left[ \left| K_m - \frac{K}{M} \right| > \sqrt{2K\ln\left(\frac{2MC}{\delta}\right)} + M\ln\left(\frac{2MC}{\delta}\right) \right] \leq \frac{\delta}{MC^M}.$$

When $K > (6 + 4\sqrt{2})M^2 \ln\left(\frac{2MC}{\delta}\right)$, we have $\sqrt{2K\ln\left(\frac{2MC}{\delta}\right)} + M\ln\left(\frac{2MC}{\delta}\right) < \frac{K}{2M}$. By a union bound over all possible hard instances $\mathcal{M}(\boldsymbol{\ell}^\star, \boldsymbol{a}^\star) \in \mathcal{C}$ and all indices $m \in [M]$, the following event happens with probability at least $1 - \delta$:

$$\mathcal{E} := \left\{ K_m \geq \frac{K}{2M} \text{ for all } \mathcal{M}(\boldsymbol{\ell}^\star, \boldsymbol{a}^\star) \in \mathcal{C} \text{ and } m \in [M] \right\}.$$

Now look into Equation (8), (11) and (12) of Domingues et al. (2021). For any $K' \geqslant SA$ and any fixed encoding number $m$, we have that

$$\frac{1}{C} \sum_{\ell^\star, a^\star} R(\mathcal{M}(m, \ell^\star, a^\star), \boldsymbol{\pi}(m, \boldsymbol{\ell}^\star, \boldsymbol{a}^\star), K') \geqslant \frac{1}{4\sqrt{2}} \left(1 - \frac{1}{C}\right) \sqrt{CK'} \geqslant \frac{\sqrt{CK'}}{8\sqrt{2}}, \qquad (9)$$

when set $\varepsilon = \frac{1}{2\sqrt{2}} \left(1 - \frac{1}{C}\right) \sqrt{\frac{C}{K'}}$. The desired value of $K'$ is $\frac{K}{2M}$ according to $\mathcal{E}$.

We study the cases when we use $\boldsymbol{\pi}(m, \boldsymbol{\ell}^\star, \boldsymbol{a}^\star)$ to solve $\mathcal{M}(m, \ell_m^\star, a_m^\star)$ with a target interaction episode $\overline{K}_m = \frac{K}{2M}$. The regret is $R(\mathcal{M}(m, \ell_m^\star, a_m^\star), \boldsymbol{\pi}(m, \boldsymbol{\ell}^\star, \boldsymbol{a}^\star), \frac{K}{2M})$ (this is the regret of MDPs).

- The $\frac{K}{2M}$th interaction with the $m$th MDP comes before the $K$th simulation episode. This case happens under $\mathcal{E}$. The regret of this part is denoted as $R^+$.

- Otherwise. This case happens under $\bar{\mathcal{E}}$. The regret of this part is denoted as $R^-$. Since the regret of a single episode is at most 1, we have that $R^- < \frac{\delta K}{2M}$.

Now we study the cases when we use $\boldsymbol{\pi}(m, \boldsymbol{\ell}^\star, \boldsymbol{a}^\star)$ to solve $\mathcal{M}(m, \ell_m^\star, a_m^\star)$ with a simulation episode budget $K$. The alternative regret for MDP is $\widetilde{R}(\mathcal{M}(m, \ell_m^\star, a_m^\star), \boldsymbol{\pi}(m, \boldsymbol{\ell}^\star, \boldsymbol{a}^\star), K)$.

- The $\frac{K}{2M}$th interaction with the $m$th MDP comes before the $K$th simulation episode. This case happens under $\mathcal{E}$. The regret of this part is denoted as $\widetilde{R}^+$. Since the regret of a single episode is at least 0, and in this case $K_m \geqslant \frac{K}{2M}$, we have $\widetilde{R}^+ \geqslant R^+$.

- Otherwise. This case happens under $\bar{\mathcal{E}}$. The regret of this part is denoted as $\widetilde{R}^- \geqslant 0$.

Using the connection between $R^+$ and $\widetilde{R}^+$, we have:

$$\frac{1}{C^M} \sum_{\boldsymbol{\ell}^\star, \boldsymbol{a}^\star} R(\mathcal{M}(\boldsymbol{\ell}^\star, \boldsymbol{a}^\star), \boldsymbol{\pi}, K)$$

$$= \frac{1}{C^M} \sum_{\boldsymbol{\ell}^\star, \boldsymbol{a}^\star} \sum_{m=1}^{M} \widetilde{R}(\mathcal{M}(m, \ell_m^\star, a_m^\star), \boldsymbol{\pi}(m, \boldsymbol{\ell}^\star, \boldsymbol{a}^\star), K)$$

$$\stackrel{(i)}{=} \sum_{m=1}^{M} \frac{1}{C^{M-1}} \sum_{\boldsymbol{\ell}_{-m}^\star, \boldsymbol{a}_{-m}^\star} \frac{1}{C} \sum_{\ell_m^\star, a_m^\star} \widetilde{R}(\mathcal{M}(m, \ell_m^\star, a_m^\star), \boldsymbol{\pi}(m, \boldsymbol{\ell}^\star, \boldsymbol{a}^\star), K)$$

$$\geqslant \sum_{m=1}^{M} \frac{1}{C^{M-1}} \sum_{\boldsymbol{\ell}_{-m}^\star, \boldsymbol{a}_{-m}^\star} \frac{1}{C} \sum_{\ell_m^\star, a_m^\star} \widetilde{R}^+(\mathcal{M}(m, \ell_m^\star, a_m^\star), \boldsymbol{\pi}(m, \boldsymbol{\ell}^\star, \boldsymbol{a}^\star), K)$$

$$\geqslant \sum_{m=1}^{M} \frac{1}{C^{M-1}} \sum_{\boldsymbol{\ell}_{-m}^\star, \boldsymbol{a}_{-m}^\star} \frac{1}{C} \sum_{\ell_m^\star, a_m^\star} R^+ \left(\mathcal{M}(m, \ell_m^\star, a_m^\star), \boldsymbol{\pi}(m, \boldsymbol{\ell}^\star, \boldsymbol{a}^\star), \frac{K}{2M}\right)$$

$$= \sum_{m=1}^{M} \frac{1}{C^{M-1}} \sum_{\boldsymbol{\ell}_{-m}^\star, \boldsymbol{a}_{-m}^\star} \frac{1}{C} \sum_{\ell_m^\star, a_m^\star} (R - R^-) \left(\mathcal{M}(m, \ell_m^\star, a_m^\star), \boldsymbol{\pi}(m, \boldsymbol{\ell}^\star, \boldsymbol{a}^\star), \frac{K}{2M}\right)$$

$$> \sum_{m=1}^{M} \frac{1}{C^{M-1}} \sum_{\boldsymbol{\ell}_{-m}^\star, \boldsymbol{a}_{-m}^\star} \frac{1}{C} \sum_{\ell_m^\star, a_m^\star} \left(R \left(\mathcal{M}(m, \ell_m^\star, a_m^\star), \boldsymbol{\pi}(m, \boldsymbol{\ell}^\star, \boldsymbol{a}^\star), \frac{K}{2M}\right) - \frac{\delta K}{2M}\right)$$

$$\stackrel{(ii)}{\geqslant} \sum_{m=1}^{M} \frac{1}{C^{M-1}} \sum_{\boldsymbol{\ell}_{-m}^\star, \boldsymbol{a}_{-m}^\star} \left(\frac{\sqrt{CK}}{16\sqrt{2M}} - \frac{\delta K}{2M}\right)$$

$$= \frac{\sqrt{MCK}}{16\sqrt{2}} - \frac{\delta K}{2},$$

where in (i) we use $\boldsymbol{x}_{-m}$ to denote the positions other than $m$ in $\boldsymbol{x}$; (ii) is by setting $K' = \frac{K}{2M}$ in Equation (9). Set $\delta := \frac{\sqrt{MC}}{16\sqrt{2K}}$, then we have that

$$\max_{\boldsymbol{\ell}^{\star}, \boldsymbol{a}^{\star}} R(\mathcal{M}(\boldsymbol{\ell}^{\star}, \boldsymbol{a}^{\star}), \boldsymbol{\pi}, K) \geqslant \frac{1}{C^M} \sum_{\boldsymbol{\ell}^{\star}, \boldsymbol{a}^{\star}} R(\mathcal{M}(\boldsymbol{\ell}^{\star}, \boldsymbol{a}^{\star}), \boldsymbol{\pi}, K) > \frac{\sqrt{MCK}}{32\sqrt{2}} = \Omega\left(\sqrt{MSAK}\right).$$

This holds when $K > (6 + 4\sqrt{2})M^2 \ln\left(\frac{2MC}{\delta}\right)$ and $K' \geqslant SA$. It then reduces to

$$K \geqslant \Omega(M^2 \mathsf{poly}(\log(M, S, A)) + MSA).$$

This completes the proof. $\qquad\square$