# OpenReview forum: "Horizon-Free Reinforcement Learning for Latent Markov Decision Processes"
_ICLR.cc/2023/Conference — Submitted to ICLR 2023_

### Official Review · Reviewer_6Gt3 · 2022-10-21

**Confidence:** 4
**Correctness:** 4
**Technical Novelty And Significance:** 2
**Empirical Novelty And Significance:** Not applicable
**Recommendation:** 5

**Clarity, Quality, Novelty And Reproducibility:**

The writing of the paper is very good in general, with clear expressions for lemmas and theorems. I have listed several minor problems below. The novelty seems to be a problem, see the discussion for the weakness.
 1. Page 2 (main contributions and technical novelties): is helps -- helps
 2. Page 3 (related work): hindsign -- hindsight
 3. What is $\pi^k$ in Line 5 of Algorithm 4?


**Strength And Weaknesses:**

There are two key contributions of the paper: an exploration algorithm with provable improved regret bound for LMDP when the context is revealed at the end of each episode, and a novel hard instance construction and the corresponding regret lower bound. The exploration algorithm is mainly built on the model-based and model-free horizon-free techniques proposed by previous works. The construction of the hard instance and proof of the lower bound are new, whose main idea is to split the regret into $M$ parts and show that each part has to be at least $\Omega(\sqrt{SAK/M})$.

On the other hand, I have some concerns regarding to the paper:
- When the true context is revealed at the end of each episode, one can construct the empirical estimate of the transition and reward using MLE simply, which is in sharp contrast to the partially observable setting. This perhaps leads to a very similar confidence set (both for the model and the optimal value function) as used in previous works studying horizon-free exploration. The extra $\sqrt{\Gamma}$ term in the regret seemingly comes from the union bound over any $\alpha$-vector for the optimism to hold, which is similar to the idea used in UCRL2. It appears to me that this part of the paper is mostly a direct application of existing techniques.
- I agree that the lower bound construction is novel and the proof of the lower bound requires a delicated manipulation over existing results. The question is the paper claims that it uses the symmetrization technique from TCS, but I cannot find the explanation of this technique in the proof. On the other hand, the hard instance construction seems to be somehow straight forward, by reducing to $M$ independent MDPs since the true MDP can be fully identified in the first phase.
- The most interesting story to me for the problem is the minimax regret bound, since I believe the extra $\sqrt{\Gamma}$ term comes from the union bound over $\alpha$-vectors, but we do not need a union bound overall. Could you provide any intuitions or ideas for removing this term? Or you believe the lower bound can be improved.

----------Post rebuttal-----------

Thanks for explantionations. However, I still see this work unqualified for a conference paper. The setting is very limited, and many techniques are adapted from previous works. I recommend the authors to study further on the horizon-free setting in LMDPs, for example, the hidden context setting where one has to construct an estimated belief to obtain valid samples.

**Summary Of The Paper:**

This paper studies the regret minimization problem of finite horizon Latent MDP with context in hindsight, which assumes the true context of the MDP will be revealed at the end of each episode. It proposes a sample efficient algorithm and proves the polynomial regret bound for the algorithm. The algorithm utilizes the optimistic model-based and value-based planning principle with standard high probability confidence set construction. Based on the horizon-free optimistic planning techniques, the regret of this algorithm is $\tilde{O}(\sqrt{MS\Gamma AK})$, which scales with only $\log H$. It also provides a corresponding lower bound of regret $\Omega(\sqrt{MSAK})$ with a novel symmetrization technique.


**Summary Of The Review:**

Overall, I think this paper is not good enough. The major drawback is the lack of novelty, since the paper mainly builds its theory based on direct applications of existing results except for the hard instance construction. However, I really appreciate the minimax regret bound for this problem.

---

> ### Author Response · Authors · 2022-11-05
> **Response to Reviewer 6Gt3**
>
> Thank you for your review!
>
> **Concerns:**
> - For the upper bound proof, we in deed combined the idea from UCRL2B [Fruit et al., 2020] and MVP [Zhang et al., 2021a]. However, we do not do any union bound over any alpha vector, because there is $O((S A)^H)$ many of them, and doing a union bound will incur an $O(H \log (S A))$ term in the confidence set, **prohibiting horizon-free**.
>
> - Sorry for missing it in the paper, and we are sure to add more comments in the revision. By saying symmetrization, we mean that randomly give a single component of a $M$-component system to the agent, while making it unable to distinguish between which position this component is in the whole system. Though not the same as in TCS, the idea to **reduce $M$ components to $1$ component shares the same spirit as that of symmetrization** (see Section 1.1.1 of Phillips et al. [2012]).
>
> - The extra $\sqrt{\Gamma}$ actually comes from bounding the $L_1$-norm of the error between the estimated transition $\hat{P}$ and the real transition $P$ (see Lemma 10). In MDPs, people can avoid bounding this $L_1$-norm in the leading order term because they can aim to bound $ (\hat{P} - P) V^\star $ for **only** the $V^\star$. In LMDPs, due to the lack of Bellman-optimality (see our reply to all), there is no such unique $\alpha^\star$, so we must bound $ (\hat{P} - P) \alpha $ for **any** $\alpha$ by bounding the $L_1$-norm of error. Currently, we have no idea about how to **avoid Bellman-optimality to prove optimal regret for MDPs**, let alone for LMDPs.
>
> **Minor problems:** Thanks for pointing them out! **3.** There is no $\pi^k$ in Algorithm 4. The $m'$ in Line 4 should be $m^k$. Here $m^k$ is the random context for the $k$-th episode in the simulator.

---

### Official Review · Reviewer_vZUE · 2022-10-24

**Confidence:** 3
**Correctness:** 4
**Technical Novelty And Significance:** 2
**Empirical Novelty And Significance:** Not applicable
**Recommendation:** 3

**Clarity, Quality, Novelty And Reproducibility:**

The paper is very clear, and well written. The results are new and I believe correct. The methodology is very classical.

There is no simulation to reproduce.

**Strength And Weaknesses:**

This paper is very nicely done but has (too me) limited impact:
- it is precise, well-done, and contains an upper bound that (almost) matches the lower bound.
- it contains an algorithm that has no practical application, and that cannot be implemented. It is not very original and (to me) oversold.

To give more details:
- even if I did not check the proofs, I do think that the results hold because the techniques are very similar to many classical in the theoretical reinforcement learning literature. They are so close that I cannot really see what is special about latent MDPs in this settign: the confidence bounds are the same as for MDPs (because the latent variable m can be observed at the end of the episode), only the lower bound is different, but is a mere adaptation of the model of Domingues et al (2021).
- The proposed algorithm is impractical for several reasons, and in particular:
  - the number of possible values for "h" is exponential in H.
  - the authors claim that one could restrict oneself to a subclass of policies, for instance Markovian policies but:
    - it is not clear if an optimal policy is easy to compute for LMDP.
    - the authors sells "extended value iteration" but this would not work for many Markovian policies.
- Some claims are too strong for me:
  - The authors claim that this "symmetric technique" is very new -> having variant of the same problem seems very standards and not very close to the symmetric technique in TCS.
  - the claim that "the lower bound matches for \Gamma=2" is clearly true but not very informative. It does not mean anything for larger \Gamma.


**Summary Of The Paper:**

This paper studies episodic learning algorithms for a variant of POMDP, called LMDP. In such a setting, there are M different MDPs and a new MDP (say m) is drawn at the beginning of each episode. The learner only observes m at the end of the episode and uses to learn. The authors derive both an upper bound and a lower bound for their problem, that are very close.


**Summary Of The Review:**

This paper is a nice treatment of a purely theoretical problem. I do not find it very original nor applicable.

---

> ### Author Response · Authors · 2022-11-05
> **Response to Reviewer vZUE**
>
> Thank you for your review!
>
> - For the upper bound proof, we in deed combined the idea from UCRL2B [Fruit et al., 2020] and MVP [Zhang et al., 2021a]. While for the lower bound, the idea is to **introduce the idea of symmetrization to the RL field**. We used the model of Domingues et al. [2021] as a **blackbox component**, together with a "**simulator** algorithm" (Algorithm 4) to scale up a factor of $\sqrt{M}$. However, if we directly adapt the construction of Domingues et al. [2021] to an LMDP instance distribution, it would **fail to contain any $M$ in the regret lower bound**. This is because the lower bound proofs for MDPs all contain a KL divergence, which is tight when the distribution support is $2$, but not tight when it is $M$. As a result, the $M$ terms will cancel out in the final regret lower bound. The idea of symmetrization is **useful for the (possible) future proof** of a lower bound containing $\sqrt{\Gamma}$: if one can prove a lower bound for $M=2$ case, then the generalization to arbitrary $M$ just follows our proof in Appendix B.4.
>
> - As we mentioned in Section 2, regarding the policy solvers (Algorithm 2 and 3), it is unavoidable to have an exponential time complexity as shown by Steimle et al. [2021]. For time-independent policies, it is exponential as well [Littman, 1994]. We claim that the result can be restricted to any subclass of policies to show the **flexibility of our algorithmic framework and analysis**. To make the algorithm practical, we can use oracles to approximate the optimal policy. This setting was used by Kwon et al. [2021]. Suppose the oracle for Algorithm 3 returns a policy $\pi'$ such that $\tilde{V}^{\pi'} \ge C \max_{\pi \in \Pi} \tilde{V}^\pi$, then the regret would be defined as $\sum_{k = 1}^K (C V^\star - V^k)$. In this case, **our analysis still holds, with a $C$ timed to the regret**.
>
> -
>     - By saying symmetrization, we mean that randomly give a single component of a $M$-component system to the agent, while making it unable to distinguish between which position this component is in the whole system. Though not the same as in TCS, the idea to **reduce $M$ components to $1$ component shares the same spirit as that of symmetrization** (see Section 1.1.1 of Phillips et al. [2012]). To the best of our knowledge, this idea has not been used in RL to prove lower bounds.
>
>     - This claim is to show that in part that our algorithm is auto-adaptive. It does not require prior knowledge of $\Gamma$ to adapt to this case.

---

### Official Review · Reviewer_jzYm · 2022-10-28

**Confidence:** 3
**Clarity, Quality, Novelty And Reproducibility:** The contributions seem well explained.
**Correctness:** 4
**Technical Novelty And Significance:** 3
**Empirical Novelty And Significance:** 3
**Recommendation:** 6

**Strength And Weaknesses:**

The results seem to improve the state of the art, both for the improvement in terms of H and S.

The paper assumes stationary MDP which is independent of epoch in the episode. The key aspect of episodic setup is that the transitions and the reward function could depend on h. How do the results extend in this case? I believe that an extra H dependence will come in the result. Extending to this setup can have additional nice results for the paper.

**Summary Of The Paper:**

This paper studies regret minimization for reinforcement learning (RL) in Latent Markov Decision Processes (LMDPs) with context in hindsight. Episodic undiscounted case is studied, and regret guarantees are found. The regret bound only scales logarithmically with the planning horizon. Further, the state term dependence is reduced.

**Summary Of The Review:**

This paper studies regret minimization for reinforcement learning (RL) in Latent Markov Decision Processes (LMDPs) with context in hindsight. Episodic undiscounted case is studied, and regret guarantees are found. The regret bound only scales logarithmically with the planning horizon. Further, the state term dependence is reduced.

---

> ### Author Response · Authors · 2022-11-05
> **Response to Reviewer jzYm**
>
> Thank you for your review!
>
> In this paper, we assumed stationary (**time-homogenous**) LMDPs. Just as you said, for **time-inhomogenous** LMDPs, there would be a $\sqrt{H}$ timed to the current regret. This is analogous to MDP cases, where people say horizon-free when the regret for **time-homogenous** MDPs has no $H$ except for logarithm terms. We will add this point as a remark in the revision.

---

### Author Response · Authors · 2022-11-05
**Technical difficulties in LMDPs**

Bellman-optimality ($V^\star(s)=\max_a Q^\star(s, a)$) of MDPs does not hold in LMDPs. Though the value and $Q$ functions of LMDP satisfy this, the **alpha vectors** do not. This directly leads to the **nonuniqueness of optimal alpha vectors**.
For example: $M=2, S=1, A=2, H=1$ (in fact a latent bandit, only for illustration); $w_1=w_2=1/2$; $R_1 (s, \cdot)=[1, 0], R_2 (s, \cdot)=[0, 1]$. Then both $a_1$ and $a_2$ can be the optimal action, but the alpha vectors are different in both cases.

To make things worse, even if the optimal alpha vector is unique, the Bellman-optimality still does not hold.
For example: $M=2, S=1, A=3, H=1$; $w_1=w_2=1/2$; $R_1 (s, \cdot)=[1, 0, 2/3],R_2 (s, \cdot)=[0, 1, 2/3]$. The unique optimal action is $a_3$, but it is not the optimal action when focusing on any MDP.

Due to this, **the optimistic alpha vectors cannot pointwise converge to an optimal alpha vector**. The optimism in LMDPs ($V^k (s_1) \ge V^\star (s_1)$) is thus much weaker than that in MDPs ($V_h^k (s) \ge V_h^\star (s)$ for any $s$ and time $h$). Even to achieve this, we need to impose a $\sqrt{\Gamma}$ term to the bonus. Thus we believe that **to remove $\sqrt{\Gamma}$ requires avoiding using Bellman-optimality in the proof**, which seems unrealistic even in the proof of MDP bounds.

---

### Decision · Program_Chairs · 2023-01-20

**Decision:**

Reject

**Justification For Why Not Higher Score:**

It is not so interesting, even within the narrow theoretical context.

**Justification For Why Not Lower Score:**

N/A


**Metareview: Summary, Strengths And Weaknesses:**

This paper has a clear, but small, technical contribution. Even though the paper has no technical faults, none of the reviewers, have found the paper to advance the state of the art significantly, in the sense that it does not contain either significant new results, or proof techniques. For that reason, we recommend rejection.